# Laboratory study of the collection efficiency of submicron aerosol particles by cloud droplets.
# Part II - Influence of electric charges.

## Alexis Dépée [1,2], Pascal Lemaitre[1*], Thomas Gelain[1], Marie Monier[2,3], Andrea Flossmann[2,3]

[1]    {Institut de Radioprotection et de Sûreté Nucléaire (IRSN), PSN-RES, SCA, Gif-sur-Yvette, 91192, France}

[2]    {Université Clermont Auvergne, Laboratoire de Météorologie Physique, Clermont-Ferrand, France}

[3]    {CNRS, INSU, UMR 6016, LaMP, Aubière, France}

*Correspondence to: Pascal Lemaitre (pascal.lemaitre@irsn.fr)

## ABSTRACT

A new In-Cloud Aerosol Scavenging Experiment (In-CASE) has been developed to measure the collection efficiency (CE) of submicron aerosol particles by cloud droplets. Droplets fall at their terminal velocity through a one-meter-high chamber in a laminar flow containing aerosol particles. At the bottom of the In-CASE's chamber, the droplet train is separated from the aerosol particles flow and the droplets are collected in an impaction cup whereas aerosol particles are deposited on a High Efficiency Particulate Air (HEPA) filter. The collected droplets and the filter are then analysed by fluorescence spectrometry since the aerosol particles are atomised from a sodium fluorescein salt solution ($C_{20}H_{10}Na_2O_5$). In-CASE fully controls all the parameters which affect the CE - the droplets and aerosol particles size distributions are monodispersed, the electric charges of droplets and aerosol particles are known and set, while the relative humidity is indirectly controlled via the chamber's temperature. This paper details the In-CASE setup and the dataset of 70 measurements obtained to study the impact of the electric charges on CE. For this purpose, droplets and particles charges are controlled through two charging systems developed in this work - both chargers are detailed below. The droplet charge varies from $-3.0 \times 10^4 \pm 1.4 \times 10^3$ to $+9.6 \times 10^4 \pm 4.3 \times 10^3$ elementary charges while the particle charge ranges from zero to $-90 \pm 9$ elementary charges depending on the particle radius. A droplet radius of $48.5 \pm 1.1$ µm has been considered for four particle dry radii between 100 and 250 nm while the relative humidity level during experiments is $95.1 \pm 0.2$ %. The measurements are then compared to theoretical models from literature – showing good agreement.

## INTRODUCTION

Aerosol particles (APs) are a fundamental part of the atmosphere since they act on climate and more locally on meteorology (Twomey, 1974). They are also a key topic in human health where APs are known to increase the mortality (Dockery et al., 1992). For these reasons, the processes involved in the removing of the atmospheric AP have been investigated extensively over the last decades, through theoretical works (Slinn and Hales, 1971; Beard, 1974; Slinn, 1974; Young, 1674; Grover and Beard, 1975; Grover et al., 1977; Slinn, 1977; Davenport et al., 1978; Wang et al., 1978; Flossmann, 1998; Santachiara et al., 2012; Tinsley and Zhou, 2015; Cherrier et al., 2017; Dépée et al., 2019) as well as experimental measurements in lab (Kerker and Hampl, 1974; Wang and Pruppacher, 1977; Lai et al., 1978; Barlow and Latham, 1983; Pranesha and Kamra, 1996; Vohl et al., 2007; Ladino et al., 2011; Quérel at al., 2014; Ardon-Dryer et al., 2015; Lemaitre et al., 2017; **Dépée et al., 2020**) and the environment (Volken and Schumann, 1993; Lasko et al., 2003; Chate and Pranesha, 2004; Depuydt, 2013; Laguionie et al., 2014). Far away from the source, APs are mainly scavenged through their collection by clouds and precipitations (Jaenicke, 1993) - referred as the wet deposition. Since it has been reported that the AP collection by clouds is dominated by wet deposition (Flossmann, 1998; Laguionie et al., 2014), the in-cloud AP collection remains an essential issue for the atmospheric sciences.

As previously stated in Part I of this work (**Dépée et al., 2020**) – « In most of current AP wet removal models - like DESCAM (Detailed SCAvenging Model, Flossmann, 1985) - the AP collection is described

through a microphysical parameter called "collection efficiency" (CE) which quantifies the ability of a droplet to capture the APs present in its surroundings during its fall. It is the ratio between the AP number (or mass) collected by the droplet over the AP number (or mass) within the volume swept by the droplet for a given AP radius. Another equivalent definition is the ratio of the cross-sectional area inside which the AP trajectories are collected by the droplet over the cross-sectional area of the droplet.

Many microphysical effects influence this CE and their contribution is mainly depending on the AP size. To be collected an AP has to deviate from the streamline around the falling droplet to make contact with it. The nanometric AP's trajectory is affected by the collisions with air molecules - referred as the Brownian diffusion. It results in random movement patterns (see Figure 1, A) which tend to increase the CE when the AP radius decreases. For massive APs, there is an increase of CE as they retain an inertia strong enough to deviate significantly from the streamline when it curves and to move straight toward the droplet surface - known as inertial impaction (see Figure 1, B). When considering intermediate AP size, the CE goes through a minimum value called the "Greenfield gap" (Greenfield, 1957) where the AP diffusion and inertia are weaker. In this gap, other microphysical effects can be involved to make the droplet encounter the AP like the interception for instance. It is the collection of APs following a streamline that approaches the droplet within a distance equivalent to the particle radii ($a$) - see Figure 1, C». Note that there are also thermophoretic and diffusiophoretic forces which can have an influence on the CE. These effects prevail in subsaturated air - as it is the case sometimes in clouds - and are discussed in Part I (**Dépée et al., 2020**).

Since droplets are naturally charged in clouds (Takahashi, 1973) as well as the atmospheric APs, there are electrostatic forces which can influence the AP collection. Numerous numerical studies were dedicated to the influence of the electric charges on CE – such as Grover et al. (1975), Jaworek et al. (2002), Tinsley and his group (for instance - Tinsley et al., 2006 or Tinsley and Zhou, 2015). They suggest an increase of the CE of several orders of magnitude even when the AP is weakly charged. However, the AP charge increases when the APs are radioactive (Clement and Harrison, 1992) - inducing an impact on CE even larger (Dépée et al., 2019). Thus, the AP "electroscavenging" in clouds has to be investigated, particularly for nuclear safety issues when the APs removal by clouds result from the discharge of radioactive materials from a nuclear accident. For this purpose, the modelled CEs with electrostatic forces need to be experimentally validated before the incorporation in cloud models. Especially, the analytical expression for electrostatic forces used in numerical studies (Jaworek et al., 2002 ; Tinsley et al., 2006 ; Tinsley and Zhou, 2015 ; Dépée et al., 2019) has to be confirmed by measurements.

When a droplet with a charge $Q$ approaches an AP of charge $q$, the partial influence of the AP electrostatic field on the droplet leads to the re-orientation of the water dipoles. As a result, a surface charge distribution on the droplet is created and supposed to be comparable to the one of a conductive sphere. In an electrostatic equivalent problem, the droplet can be replaced by two point charges (Jackson, 1999). One modelling the charge distribution, inside the droplet and near its surface, another for the residual droplet charge located at the droplet surface. Finally, the analytical expression of the electrostatic forces is the addition of two Coulomb forces between the AP and the two-point charges inside the droplet. The factored expression can be found in equation (10) and further details can be found in Tinsley et al. (2000). It consists of two terms. The first one is the Coulomb inverse square term which prevails in the AP collection for large enough AP electrical mobilities or electric charge products ($q \times Q$), attractive (Figure 1, D) or repulsive (Figure 1, E) depending on whether the AP charge ($q$) and the droplet charge ($Q$) have unlike or like signs. The second term is referred as the short-range attractive term and dominates for weak electric charge products or for small AP electrical mobilities (Figure 1, F) and is always attractive (due to the charge distribution at the droplet surface with opposite sign to the AP charge). A detailed study of their contribution can be found in Tinsley and Zhou (2015) or Dépée et al. (2019).

Several laboratory studies investigated the influence of the electric charges on the CE (Beard, 1974; Wang and Pruppacher, 1977; Lai et al., 1978; Barlow and Latham, 1983; Wang et al., 1983; Byrne and Jennings, 1993; Lemaitre et al., 2020) – summarised in Table 1. However, most of these works have faced difficulties in controlling all parameters impacting the CE. For instance, Beard (1974) did not measure the AP charge; Lai et al. (1978) used a polydispersed AP size distribution, the relative humidity level was not provided and the terminal velocity of the droplets was not reached ; Barlow and Latham (1983) used a polydispersed AP size distribution and the relative humidity level significantly varied from 50 to 70 % in their measurements; in the work of Byrne and Jeannings (1993) the droplet velocity does not reach the terminal velocity; the relative humidity measured in Lemaitre et al. (2020) varied from 27 and 37 %. For these reasons, it is really difficult to find comparable CE measurements in the literature as Barlow and Latham (1983) concluded after highlighting a

discrepancy of few orders of magnitude between all these authors. Nevertheless, Wang and Pruppacher (1977) and Wang et al. (1983) succeeded in controlling the charges and the sizes (as well as the relative humidity for Wang and Pruppacher (1977) but they considered only unlike signs between both droplets and APs. In their study, Lemaitre et al. (2020) did not observed any influence of electric charges on CE since for the low relative humidity level and the large droplet radius considered, the diffusiophoretis and thermophoresis dominated the AP collection.

Thus, only the Coulomb inverse square term in the analytical expression of the electrostatic forces can be documented whereas the contribution of the short-range attractive term has not been experimentally verified until now.

Table 1 Laboratory studies focused on the influence of electric charges on the CE. Charges are presented in number of elementary charges.

| Parameter / Study | Droplet radius (µm) | AP radius (µm) | Droplet charge (\|e\|) | AP charge (\|e\|) | Relative humidity (%) | Terminal velocity |
|---|---|---|---|---|---|---|
| Beard (1974) | 200 - 425 | Monodisperse 0.35-0.44 ± 0.04 | $10^4$ - $10^6$ | **Not measured** | 99 | Reached |
| Wang and Pruppacher (1977) | 170 - 340 | Monodisperse 0.25 ± 0.03 | $10^6$ | 9 & 15 ± 2 | 23 ± 2 | Reached |
| Lai et al. (1978) | 620 & 820 | **Polydisperse** 0.15 - 0.45 | $10^7$ - $10^8$ | **Not measured** | **Neither controlled nor measured** | **Not reached** |
| Barlow and Latham (1983) | 270 - 600 | **Polydisperse** 0.2 - 1 | $10^4$ - $10^7$ | Neutralised | **Uncontrolled but measured** 50 - 70 | Reached |
| Wang et al. (1983) | 69 - 250 | Monodisperse 0.038 - 0.1 | $10^7$ - $10^8$ | **1 - 13.5** | **Uncontrolled** | Reached |
| Byrne and Jennings (1993) | 400 - 500 | Monodisperse 0.35 - 0.88 | $10^5$ - $10^8$ | 360 - 750 | 57 | **Not reached** |
| Lemaitre et al. (2020) | 1275 | **Polydisperse** 0.15 and 0.25 | $10^5$ - $10^8$ | Neutralised and measured (0 ± 0.1) | **Uncontrolled but measured** 27 - 37 | Reached |

The purpose of this study is to overcome this lack of data by providing new CE measurements for weakly and strongly droplets and APs charges with both negative and positive charge products, to quantify the effect of the short-range attractive term on the CE since its contribution was previously predicted by modelling (Tinsley and Zhou, 2015; Dépée et al., 2019).

Thus, a novel experiment has been designed to study the influence of electric charges on the CE which is presented in this paper. Note that this experiment was also used to study the influence of relative humidity which is the object of the companion paper: Part I **(Dépée et al., 2020)**.

The first part of the paper describes the experimental setup. Afterwards, the method to evaluate the CE and the uncertainties are detailed. Then, the measurements are presented and confronted with the prediction of Kraemer and Johnstone (1955) and the Lagrangian model of Dépée et al. (2019). Finally, this work concludes with the experimental validation of the Dépée et al. (2019) model

and a necessary incorporation of the modelled CEs in cloud models, pollution models, climate models,
and so forth, to study the "electroscavenging".

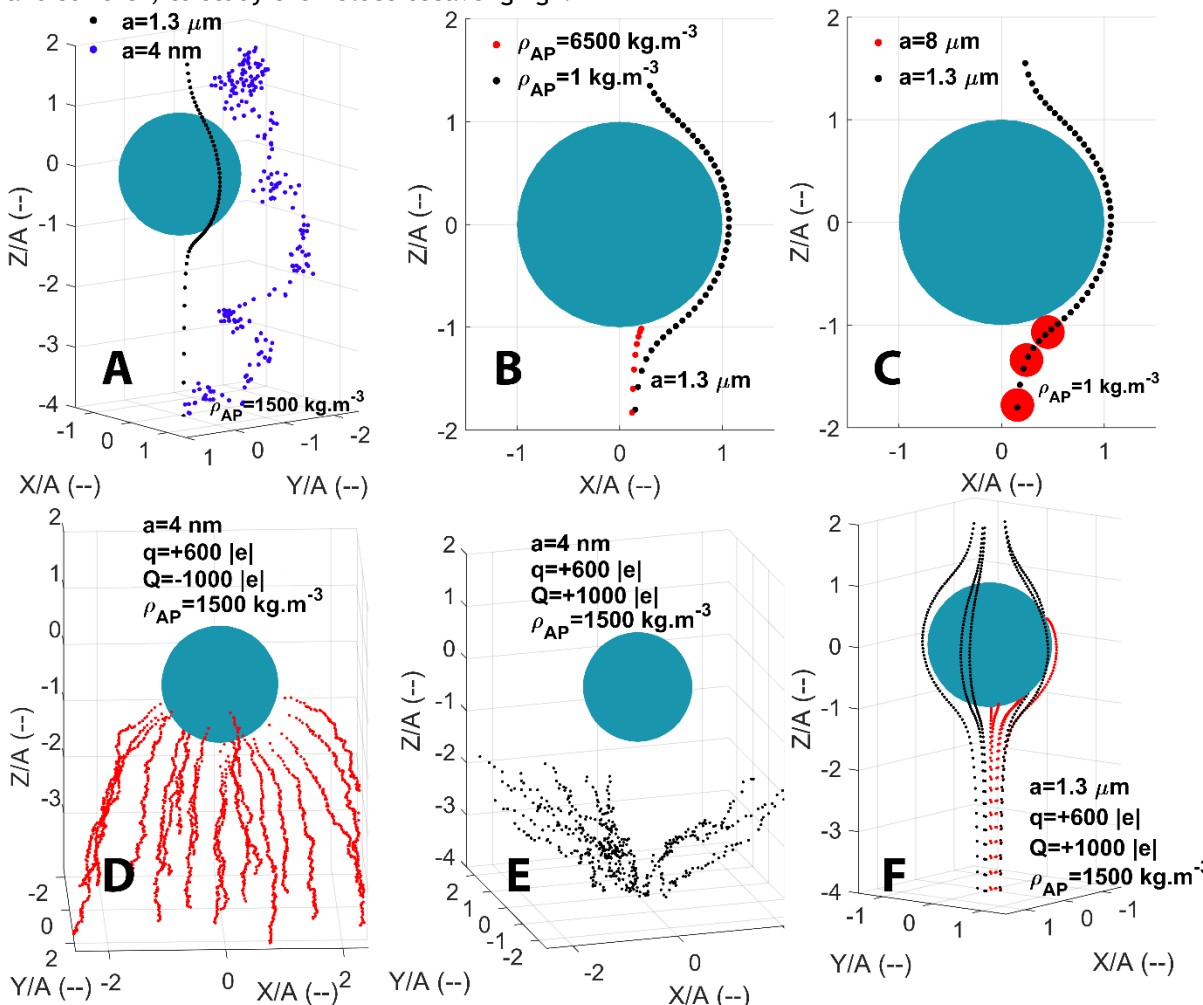

Figure 1 APs trajectories computed with the extended Dépée et al. (2019) model for a 50 µm droplet
radius ($A$) and AP with various radii ($a$) and densities ($\rho_{AP}$). The air temperature ($T_{air}$) and the air
pressure ($P_{air}$) are -17°C and 540 hPa, respectively. The panels indicate the effects of Brownian
motion (A), inertial impact (B), interception (C), electrostatic forces with attractive (D) and repulsive
(E, F) Coulomb forces. Red trajectories result in an AP collection. The droplet ($Q$) and AP ($q$) charges
are labelled. In Figures 1 B to F - the red trajectories result in an AP collection. Adapted from Part I
(**Dépée et al., 2020**).

**1      EXPERIMENTAL SETUP**
**1.1      Overview**
Figure 2 shows the In-Cloud Aerosol Scavenging Experiment (In-CASE) which has been built to study
the influence of the electric charges on the CE. Droplets fall at their terminal velocity (≈25 cm/s)
into a chamber through an AP flow of 1.5 l/min. The flow velocity is 1.3 cm/s and the AP transfer
time in the collision chamber is almost 80 s. The In-CASE's chamber is subdivided into 3 parts - the
injection head where droplets and APs are inserted; the collision chamber where droplets and APs
interact with each other; the aerodynamic separator set at the bottom's chamber impacts droplets
into an impaction cup while uncollected APs pass out of the chamber toward a High Efficiency
Particulate Air (HEPA) filter. For this latter stage, an Argon updraft assures that there are no AP that
settle into the droplet impaction cup. More details on the In-CASE's chamber can be found in section
1.2 of **Dépée et al. (2020)**.
APs are atomised from a sodium fluorescein salt solution ($C_{20}H_{10}Na_2O_5$). This molecule has been
used for its significant fluorescent properties, detectable at very low concentrations (down to $10^{-10}$
g/l). Once generated, the APs flow through a diffusion dryer and a portion of the flow is then directed
into a Differential Mobility Analyser (DMA; TSI 3080) to select APs following their electrical mobilities
whereas the overflow ends in an exhaust (black, Figure 2). At the DMA's outlet, the AP size
distribution is assumed to be monodispersed (discussed in section 2.1). Thereafter, APs are
electrically charged by a custom-designed field charger (section 1.4). Since the optimised AP flowrate
in the charger is 1.5 l/min and the maximum AP flowrate in the DMA was 1.2 l/min during the
experiments, a clean air flowrate is added at the charger's inlet. Before the AP injection in the In-
CASE's chamber, the flow is humidified to ensure a high relative humidity level inside the collision
chamber (section 1.2). Thus, the hygroscopicity of the sodium fluorescein salt is considered during
the experiments (see section 2.2). Before the AP collection on the HEPA filter, the APs flow through
a low-energy X-ray neutraliser (< 9.5 keV, TSI 3088) to eliminate charge accumulation on this filter
leading to AP deposition on the metallic walls of the filter holder. Note that the pipes are anti-static
and connected to the ground (as well as the collision chamber) so there is no accumulation charge
before the HEPA filter due to AP deposition.
Droplets are generated with a piezoelectric injector provided by Microfab (MJ-ABL-01 model) with an
internal diameter of 150 µm - at 25 Hz to prevent droplets from coalescencing. The generator is set
in a housing made with a 3D printer which is located in the injection head (Figure 5, Right). An
electrostatic inductor is also placed in the housing to charge droplets (section 1.5). The droplet size
is measured during experiments by optical shadowgraphy (with a strobe and a camera, brown color
in Figure 2) through two opposite windows in the injection head. Further details can be found in
section 1.4 of **Dépée et al. (2020)** but note that the size distributions of the droplets generated by
the piezoelectric injector are considered monodispersed since the droplet size dispersion is very low
$(\sigma \sim 1\%)$.

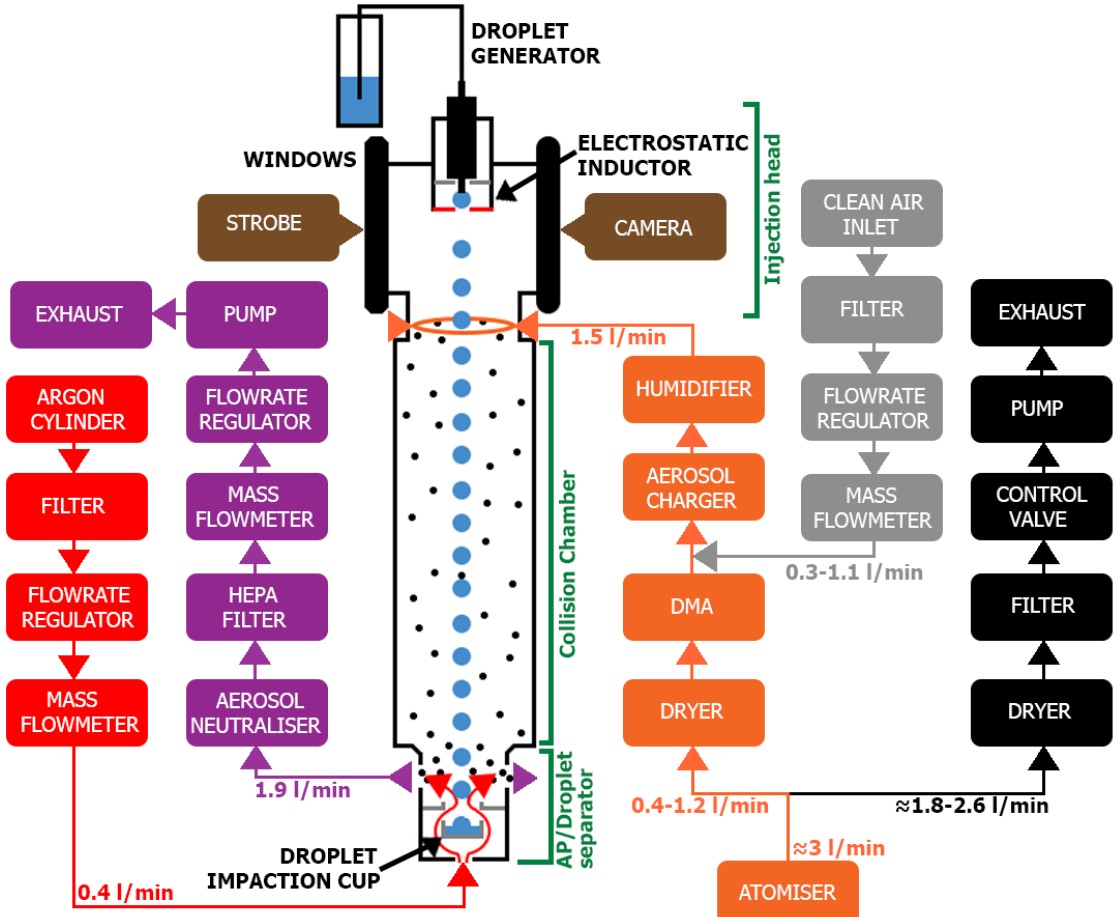

Figure 2 In-CASE setup to study the electric charges' influence - adapted from Part I (**Dépée et al.,
2020**). Colors represent different functions. Red – upward argon flow against AP pollution in
the droplet impaction cup. Purple – AP (and Argon) evacuation toward the HEPA filter after
neutralisation. Orange – AP, generation, selection and charging. Black – surplus evacuation
and DMA flowrate control. Grey – clean air adding for a constant flowrate at the AP charger's
inlet. Brown – droplet radius measurement.
**1.2      Thermodynamic conditions in the In-CASE's chamber**
Thermodynamic conditions were set as constant as possible during experiments to get comparable
CE measurements. The pressure in the In-CASE's chamber was one atmosphere and the mean
temperature for the campaign presented in this paper was 1.08 ± 0.12 °C. As referred in section 1.2
of **Dépée et al. (2020)**, the chamber's temperature is controlled through a cooling system which
indirectly sets the relative humidity level in the chamber. Here, the temperature of the pure water
in the humidifier placed before the In-CASE's chamber (Figure 2) was increased to get a mean relative
humidity level in the chamber of 95.1 ± 0.2 %. Note that this relative humidity level was the maximum
which could be reached with In-CASE. In this way, the contribution of the thermophoretic and the
diffusiophoretic effects in the CE measurements were reduced as much as possible.

## 1.3    Droplet evaporation

The droplet evaporation was theoretically evaluated through the section 13.2 of Pruppacher and Klett
(1997).   The terminal velocity of the droplet ($U_{A,\infty} \approx 25$ cm/s) is computed from Beard (1976)
meanwhile the droplet residence time in the collision chamber ($\approx 4$ s) is deduced from the changes in
droplet radius and terminal velocity. For a relative humidity level of 95 %, it was found that the
droplet radius decreases by less than 0.3 % from the droplet generation to the bottom of the collision
chamber. Thus, the droplet evaporation in the In-CASE collision chamber was neglected for the
discussions below.

## 1.4    AP charging

APs are electrically charged by passing through a custom-designed field charger adapted from Unger
et al. (2004). The scaled geometry is presented in Figure 3. This charger is based on a system of
electric discharges produced between a high potential tungsten wire and a grounded cylinder. A
metallic converging portion is used at the charger's outlet to trap ions and ensure only charged APs
can leave the charger. A Teflon ball (Ø=1 mm) is set at the end of the tungsten wire to ensure there
is no point effect between the wire and the ion trap. A large number of ions are then created and
migrate between the two centimeters interelectrode space along the electric field lines. Finally, the
APs flow through them and are charged by ion attachment.

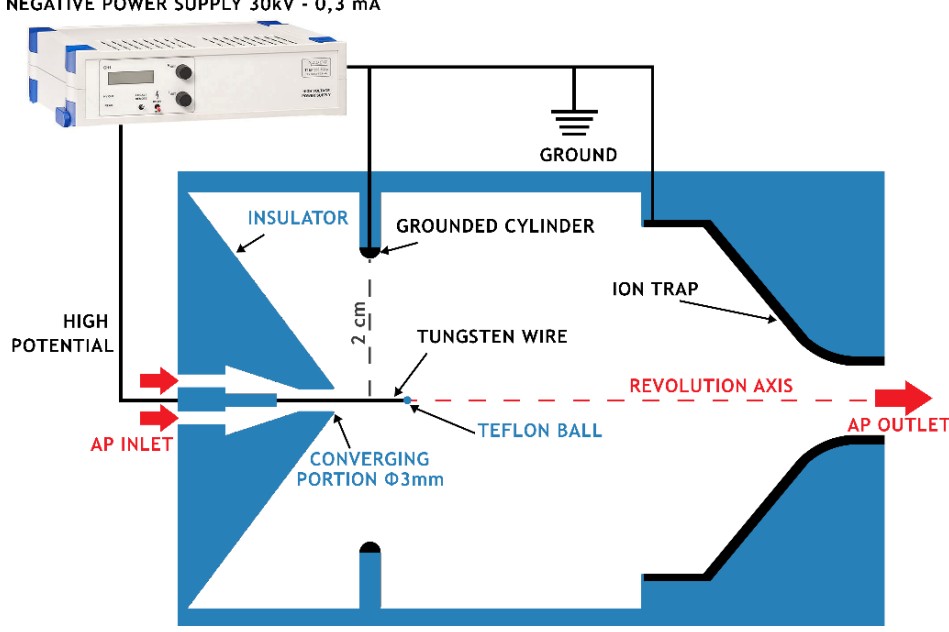

Figure 3 Geometry of the home-made AP charger based on Unger et al. (2004) (at scale).

The charging relationships of the charger used during all experiments are presented in Figure 4. They
provide the mean electric AP charge related to the potential at the tungsten wire for the 4 AP radii
considered here. It results from *ex situ* experiments which are detailed in Appendix A. Note that APs
are negatively charged through the discharge regime used (negative Trichel regime) and there is an
electric potential where the AP charge saturates which is typical for field chargers (Pauthenier and
Moreau-Hanot, 1932). These results were performed at an AP flowrate of 1.5 l/min which was found
to maximise the AP penetration inside the charger and consequently the AP concentration inside the
In-CASE's chamber. Penetration tests - not presented in this study - were deduced by varying the AP
flowrate in the setup detailed in Appendix A.

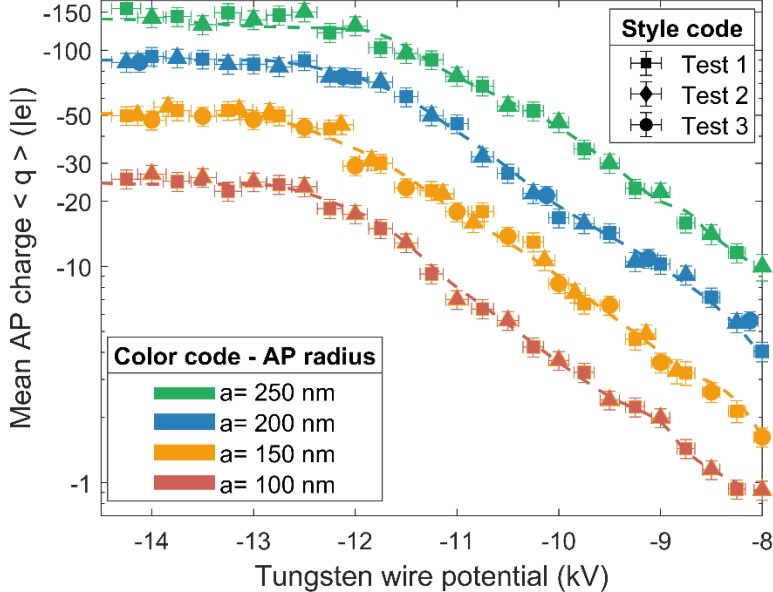

Figure 4 AP charging relationships used during experiments for the 4 AP radii ($a$) considered. Error
bars represent the standard deviations in measurements.
**1.5    Droplet charging**
The droplets charge is controlled through an electrostatic inductor adapted from Reischl et al. (1977).
Two parallel metallic plates are set at the droplet generator's nozzle (Figure 5, left) - one plate
connected to the neutral potential and the other one to a potential referred as $V_{ind}$. It induces an
electric field ($E_{ind} \sim 10^2$-$10^3$ V/m) at the nozzle. Since sodium chloride is added to the pure water that
feeds the piezoelectric droplet generator, this electric field can selectively attract negative or
positive ions toward the nozzle where the droplet is formed, according to its sign. If $V_{ind}$ is negative,
the positive sodium ions ($Na^+$) migrate toward the nozzle and the negative chloride ions ($Cl^-$) are
repulsed from the nozzle and inversely if the potential is positive. Furthermore, the amplitude of the
electric field ($E_{ind}$) sets the ion quantity in the droplet. Note that the sodium chloride concentration
has no impact on the induced droplet charge if the ion number is large enough for the entire
experiment time (Reischl et al., 1977) - 3.3 g/l was used here.

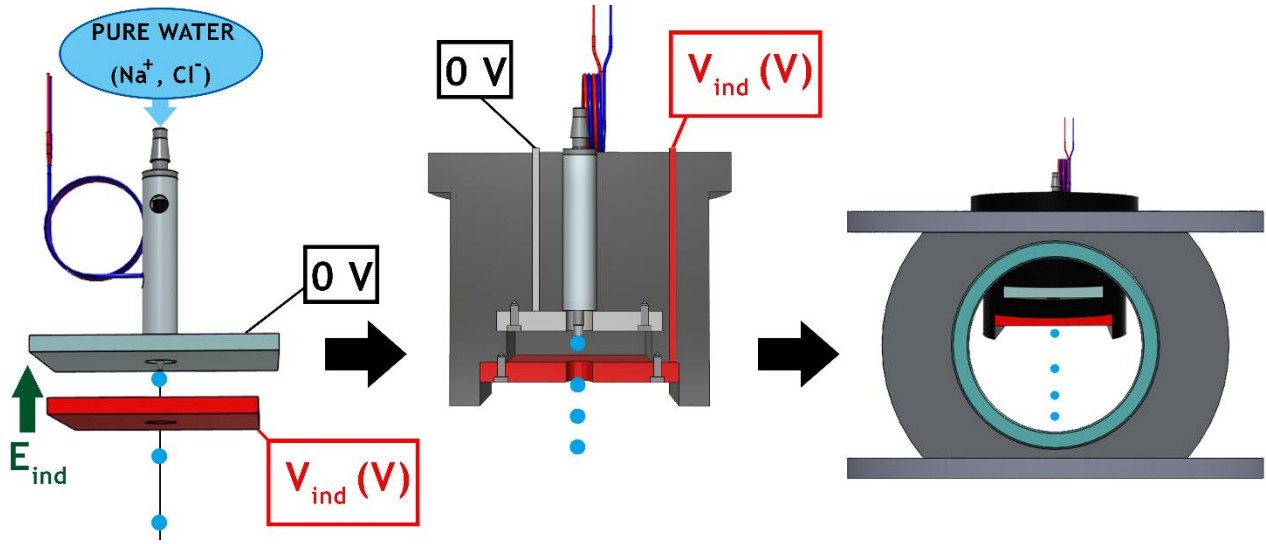


Figure 5 (**Left**) Electrostatic inductor below the piezoelectric droplet generator. (**Center**) Cross section of the housing made with a 3D printer that hold the droplet generator. (**Right**) Injection head at the top of the In-CASE's chamber containing the housing made with a 3D printer.

A method to evaluate the droplet charge was developed in this study and is detailed in Appendix B. In Figure 6, the resulting charging relationship of the electrostatic inductor is presented. It gives the droplet charge ($Q$) as a function of the electrostatic inductor potential ($V_{ind}$). We note that the droplet generator produces highly electrically charged droplets since the droplet charge is evaluated to about -8,400 elementary charges, for a zero potential at the inductor plate ($V_{ind}$= 0 V). This is in line with Ardon-Dryer et al. (2015) which used a similar generator and measured up to $10^4$ elementary charges on the generated droplets. Finally, this charging relationship is used during experiments to positively or negatively set the droplet charges. The electrostatic inductor and the droplet generator are placed into a housing made with a 3D printer (Figure 5, center), this latter being placed in the injection head at the top of the In-CASE's chamber (Figure 5, right).

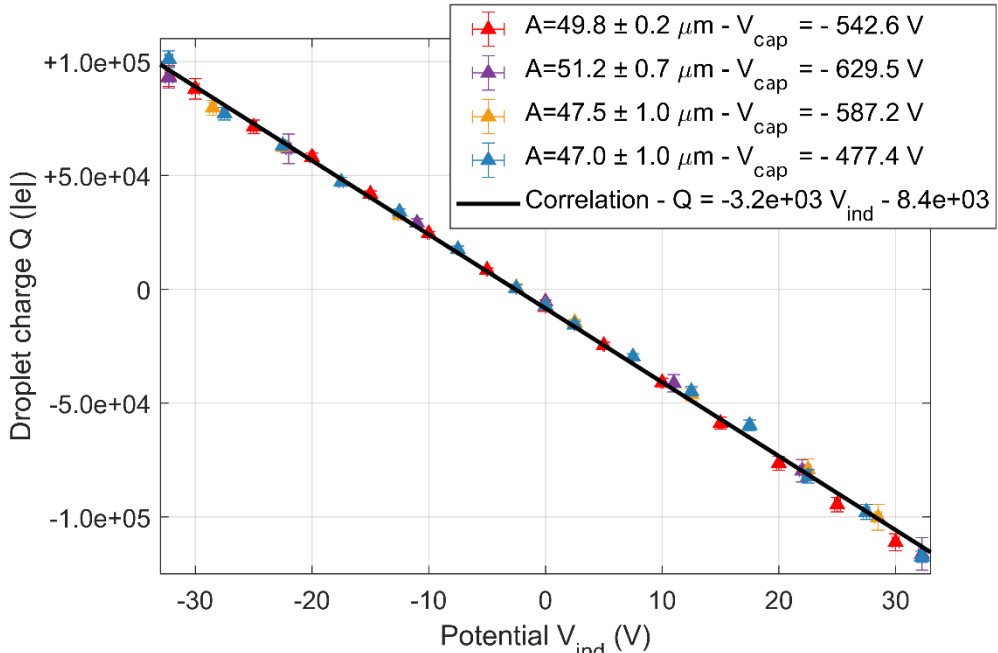

Figure 6 Droplet charge versus electrostatic inductor voltage. The colours identify different tests performed to ensure there is no droplet charge modification over time and manipulations. Error bars represent the standard deviation of the droplet radii evaluated. The parameter $V_{cap}$ is the potential at the capacitor to measure the droplet charge. Note that the radius of the droplet is deduced from the reversed Beard (1976) model and its terminal velocity – this latter being measured by optical shadowgraphy. See Appendix B for more further details.

## 2    DATA ANALYSIS

### 2.1    Assumption of a monodispersed AP size distribution

As a reminder, APs flow through a DMA (Figure 2) to select APs following their electrical mobility. As explained in section 2.2 of the Part I (**Dépée et al., 2020**), several AP radii can actually be selected depending on their elementary charges given that they have the same electrical mobility. For example, with a selected AP radius of 100 nm at the DMA (considering one elementary charge on it), the doubly-charged AP radii of 161.8 nm will also be selected.

Sometimes, the multiple charged APs cannot be neglected in the AP flow at the DMA's outlet. The CE deduction is then more difficult (**Dépée et al., 2020**). Here, the aerodynamic impactor at the DMA's inlet as well as the AP flowrate in the DMA were optimised to prevent double (and greater) charged AP from being selected by the DMA. Indeed, the cut-off radius of the impactor at the DMA's inlet (referred as $D_{50\%}/2$) - which is the radius where 50 % of the APs are impacted - is small enough

compared to the double charged AP radius. This is summarised in Table 2 for all selected AP radii
used in the experiments. Thus, the AP size distribution at the DMA's outlet is assumed to be
monodispersed and the CE is deduced as follows.

Table 2 AP selection parameters

| Selected dry AP radius by the DMA (single charged) | Double charged dry AP radius with the same electrical mobility | AP flowrate in the DMA | Cut-off radius of the impactor at the DMA's inlet ($D_{50\%}/2$) |
|---|---|---|---|
| 100 $nm$ | 161.8 $nm$ | 1.2 $l/min$ | 140 $nm$ |
| 150 $nm$ | 253.7 $nm$ | 1.0 $l/min$ | 157 $nm$ |
| 200 $nm$ | 348.3 $nm$ | 0.6 $l/min$ | 213 $nm$ |
| 250 $nm$ | 444.3 $nm$ | 0.4 $l/min$ | 268.5 $nm$ |

## 2.2 Collection efficiency definition

The collection efficiency ($CE$) is calculated from the equation (1):

$$CE(a, A, q, Q, HR) = \frac{m_{AP,d}}{m_{AP,available}}$$

(1)

Where $m_{AP,d}$ is the AP mass collected by all droplets which is directly measured by fluorescence
spectrometry analysis of the droplets collected in the impaction cup, which is located at the bottom
of the In-CASE chamber (Figure 4 of **Dépée et al., 2020**). $m_{AP,available}$ is the mass of available APs in
the volume swept by all droplets - evaluated with the equation (2) :

$$m_{AP,available} = \pi(A + GroF(RH) \times a)^2 \times F_d \times \Delta t \times H_{eff} \times C_{m,AP}$$

(2)

With $A$ - the droplet radius, $F_d$ - the droplet generation frequency, $\Delta t$ - the experiment duration (from
3 to 6 hours), $a$ - the AP dry radius, $C_{m,AP}$ – the mean AP mass concentration in the In-CASE collision
chamber, $GroF$ - the growth factor of AP depending on the relative humidity ($RH$). This latter
characterises the hygroscopicity of the sodium fluorescein salt - further details related to its
evaluation can be found in section 1.2.3.3 of **Dépée et al. (2020)**. $H_{eff}$ is the effective height of
interaction between APs and droplets calculated with the equation (3):

$$H_{eff} = \frac{U_{A,\infty}}{U_{A,\infty} + V_Q} H_{In-CASE}$$

(3)

With the AP flow velocity ($V_Q$) equal to 1.3 cm/s (for an AP flowrate of 1.5 l/min), the droplet
terminal velocity ($U_{A,\infty}$) assumed to be equal to 25 cm/s and the In-CASE collision chamber's height
($H_{In-CASE}$) of 1 meter.

In equation (2), the mean AP mass concentration ($C_{m,AP}$) in the In-CASE collision chamber is evaluated
from the fluorescence spectrometry analysis of the HEPA filter. It is given by the equation (4) where
$Q_{In-CASE,c}$ is the AP flowrate going through the In-CASE collision chamber and $m_{AP,tot}$ is the total AP
mass on the HEPA filter at the end of the experiment.

$$C_{m,AP} = \left(1 + \frac{1 - P_{InCASE,a,q}}{2}\right) \frac{m_{AP,tot}}{\Delta t \times Q_{In-CASE,c}}$$

(4)

The mean AP mass concentration is corrected considering the penetration ($P_{InCASE,a,q}$) in the collision
chamber which depends on the AP radius ($a$) and charge ($q$). This parameter was estimated during
*ex situ* experiments where the setup was the same as Figure 2, the only difference being a
Condensation Particle Counter (CPC) positioned behind the AP neutraliser and the AP charger to
measure two AP number concentrations, $n_1$ and $n_2$, respectively. The penetration is then defined as
concentration $n_1$ over concentration $n_2$. Thus, the measured penetration accounts for the AP

deposition due to electrostatic forces on the wall of the collision chamber as well as in the pipes from the AP charger to the HEPA filter and the humidifier (Figure 2). The measured penetrations are presented in Table 3. It is observed that the penetration decreases when the AP charges ($q$) increase and the AP radius ($a$) decreases since the electrical mobility of APs is larger. During experiments, the AP number concentration ranged from $3 \times 10^4$ cm$^{-3}$ (for $a$=100 nm and $= -10 \pm 1\ |e|$) to $2 \times 10^3$ cm$^{-3}$ (for $a$=250 nm and $q = -90 \pm 9\ |e|$). As a reminder, the pipes are anti-static and connected to the ground (as well as the collision chamber) so there is no charge accumulation due to AP deposition during experiments. Thus, the penetrations presented in Table 3 are assumed to be constant over time. Note that the AP deposition was neglected in Part I (**Dépée et al., 2020**) since the penetration was almost 100 % when the APs are neutralised.

Table 3 Measured penetration for the experimental conditions.

| Dry AP radius ($a$) | AP charge ($q$) | Penetration ($P_{InCASE,a,q}$) |
|---|---|---|
| 100 $nm$ | $-10 \pm 1\ |e|$ | 94.7 % |
| | $-20 \pm 2\ |e|$ | 86.0 % |
| 150 $nm$ | $-11 \pm 1\ |e|$ | 96.5 % |
| | $-30 \pm 3\ |e|$ | 86.2 % |
| 200 $nm$ | $-10 \pm 1\ |e|$ | 97.0 % |
| | $-34 \pm 3\ |e|$ | 88.8 % |
| | $-71 \pm 7\ |e|$ | 78.2 % |
| 250 $nm$ | $-22 \pm 2\ |e|$ | 94.1 % |
| | $-52 \pm 5\ |e|$ | 89.6 % |
| | $-90 \pm 9\ |e|$ | 81.8 % |

## 2.3 Uncertainties

The relative CE uncertainty ($u_{CE}$) is calculated following Lira (2003) and presented in equation (5):

$$u_{CE} = \sqrt{u_A{}^2 + u_{H_{eff}}{}^2 + u_{N_d}{}^2 + u_{m_{AP,d}}{}^2 + u_{C_{m,AP}}{}^2} \tag{5}$$

where the relative uncertainties are related to the droplet radius ($u_A \approx 1\%$), the effective height of interaction between droplets and APs ($u_{H_{eff}} \approx 4\%$), the number of injected droplets during the experiment ($u_{N_d} \approx 2\%$), the measured AP mass in the droplet impaction cup ($u_{m_{AP,d}}$) and the mean AP mass concentration in the In-CASE collision chamber during the experiment ($u_{C_{m,AP}}$).

The relative uncertainty $u_{m_{AP,d}}$ is evaluated through the equation (6) :

$$u_{m_{AP,d}} = \sqrt{u_{fluorimeter}{}^2 + u_{dilution}{}^2} \tag{6}$$

where $u_{dilution}$ is the relative uncertainty of the dilution performed during the spectrometry analysis, assumed to be equal to 1 %, and $u_{fluorimeter}$ is the relative uncertainty of the fluorimeter which can be up to 30 % when the measured AP mass is close to the detection limit. The relative uncertainty of the mean AP mass concentration in the In-CASE collision chamber ($u_{C_{m,AP}}$) is calculated through the equations (7) :

$$\begin{cases} u_{C_{m,AP}} = \sqrt{u_{m_{AP,tot}}{}^2 + u_{Q_{In-CASE,c}}{}^2 + u_{\Delta t}{}^2 + u_P{}^2} \approx \sqrt{u_{m_{AP,tot}}{}^2 + u_{Q_{In-CASE,c}}{}^2 + u_P{}^2} \\ u_{m_{AP,tot}} = \sqrt{u_{fluorimeter}{}^2 + u_{dilution}{}^2} \end{cases} \tag{7}$$

$u_{m_{AP,tot}}$ is the relative uncertainty of the measured AP mass on the HEPA filter which depends on the relative uncertainties of the dilution ($u_{dilution} \approx 1\%$) and the fluorimeter ($u_{fluorimeter} \leq 30\%$) - $u_{Q_{In-CASE,c}}$ is the relative uncertainty of the AP flowrate in the collision chamber equal to 1 % - $u_{\Delta t}$ is the relative uncertainty of the experiment duration which is neglected here. More details are addressed in Part I, section 2.3 (**Dépée et al., 2020**) where the same definitions are used, except the relative uncertainty of the AP penetration in the collision chamber ($u_P$) is added here (equation (8)).

$$u_P = \frac{1 - P_{InCASE,a,q}}{2} \tag{8}$$

As mentioned in 3.2.3.2, an AP pollution independent from the experiment (pollution during the spectrometry analysis, when the droplet impaction cup is extracted at the end of experiments, etc.) remains and should be considered in equation (5). Indeed, it can significantly increase the CE measurement, especially when the measured AP mass is close to the detection limit of the fluorescence spectrometer. Considering the experiment duration (< 6 hours), this pollution is not totally negligible for CEs below $1 \times 10^{-4}$. Rather than discarding these measurements, there low uncertainty were extended down to the lower limit of the axis in Figures 7 and 10.

Also, we assume that APs have the same charge ($q$). Even if, an AP charge distribution exists, this contribution is negligible. Nevertheless, the AP charge distribution was not measured here.

## 3 RESULTS AND DISCUSSIONS

### 3.1 Extension of the Dépée et al. (2019) model

CE measurements are compared to the model of Dépée et al. (2019) which models the electrostatic forces ($F_{elec}$) between droplets and APs in the CE calculation. Since all experiments were performed in subsaturated air ($RH = 95.1 \pm 0.2$ %), the thermophoretic ($F_{th}$) and the diffusiophoretic ($F_{df}$) forces were also considered for the comparison with the model. Indeed, **Dépée et al. (2020)** showed that the contribution of these two effects is significant even though the relative humidity is close to 100 %. Thus, the Dépée et al. (2019) model is extended here by replacing the resulting velocity at the AP location ($\boldsymbol{U}_{f@p}{}^*$ in their Equation 6) by the equation (9):

$$\boldsymbol{U}_{f@AP}{}^*(t) = \boldsymbol{U}_{f@AP}(t) + \frac{\tau_{AP}}{m_{AP}} \left( \boldsymbol{F_{buoy}} + \boldsymbol{F_{df}} + \boldsymbol{F_{elec}} + \boldsymbol{F_{th}} \right) \tag{9}$$

Where $U_{f@AP}$ is the fluid velocity at the AP location, $\tau_{AP}$ the AP relaxation time and $m_{AP}$ the AP mass. The expression of the buoyancy force ($F_{buoy}$) is detailed in equation system (B.1), $F_{df}$ and $F_{th}$ in the equations (12) of **Dépée et al. (2020).** $F_{elec}$ is defined in equation (10) :

$$\boldsymbol{F_{elec}} = \frac{q^2}{4\pi\varepsilon_0 A^2} \left[ \overbrace{\left( -\frac{r^*}{(r^{*2}-1)^2} + \frac{1}{r^{*3}} \right)}^{Short-range\ attractive\ term} + \underbrace{\frac{1}{r^{*2}} \times \frac{Q}{q}}_{Coulomb\ inverse\ square\ term} \right] \boldsymbol{u_r} \tag{10}$$

With $\varepsilon_0$ - the permittivity of the free space, $\boldsymbol{u_r}$ - the unit vector in the radial direction from the droplet centre to the AP centre, $r^*$ - the distance between the AP and droplet centres, normalised by the droplet radius $A$.

Note that radioactive APs are known to get positively charged (Clement and Harrison, 1992) whereas the APs were negatively charged in this work (Figure 4), through the charging regime used in the AP charger (for integrity of the tungsten wire over time). Nevertheless, since we have the relation $F_{elec}(q, -Q) = F_{elec}(-q, Q)$ in equation (10), the CE measurements with the same $\frac{q}{Q}$ ratios are equivalent, assuming this analytical expression is validated by the measurements (see section 3.2.3).

## 3.2    Collection efficiency measurements

The CE measurements for various charges are presented in Table 4 for the 4 wet AP radii ($a_{wet}$) considered in this study. Note that the wet AP radii are the ones of the APs which grew in the collision chamber due to their hygroscopicity. During experiments, the AP radius increases by a growth factor ($GroF$) between 1.73 and 1.75 (since we actually considered the 4 mean levels of relative humidity for the 4 AP radii used in the experiments). Further details related to the calculation of the growth factor can be found in section 1.2.3.3 of Dépée et al. (2020). In Table 4, the droplet ($Q$) and AP ($q$) charges are also informed by number of elementary charges. The mean temperature was $1.08 \pm 0.12°C$ and the mean relative humidity was $95.1 \pm 0.2$ % for a droplet radius of $48.5 \pm 1.1$ µm. Note that the wet AP density depends on the one of sodium fluorescein salt and water. The equation (1) of **Dépée et al. (2020)** yielded a density of 1110 kg.m$^{-3}$. The key features of the experiments are summarised in Table 5.

Table 4 CE measurements

| $a_{wet}$ (nm) / $q$ (\|e\|) | $Q(\|e\|)$ $9.6 \times 10^4 \pm 4.3 \times 10^3$ | $3.0 \times 10^4 \pm 1.9 \times 10^3$ | $5.0 \times 10^3 \pm 8.4 \times 10^2$ | $0 \pm 6.0 \times 10^2$ | $-5.0 \times 10^3 \pm 7.7 \times 10^2$ | $-1.0 \times 10^4 \pm 8.7 \times 10^2$ | $-3.0 \times 10^4 \pm 1.4 \times 10^3$ |
|---|---|---|---|---|---|---|---|
| $175 \pm 3$, $-10 \pm 1$ | $3.91 \times 10^{-2}$ | $2.44 \times 10^{-2}$ | $3.47 \times 10^{-3}$ | $4.17 \times 10^{-3}$ | $5.58 \times 10^{-3}$ | $9.81 \times 10^{-4}$ | $2.55 \times 10^{-4}$ |
| $175 \pm 3$, $-20 \pm 2$ | $6.77 \times 10^{-2}$ | $3.47 \times 10^{-2}$ | $6.99 \times 10^{-3}$ | $5.07 \times 10^{-3}$ | $4.25 \times 10^{-3}$ | $9.17 \times 10^{-4}$ | $4.12 \times 10^{-5}$ |
| $260 \pm 3$, $-11 \pm 1$ | $2.41 \times 10^{-2}$ | $1.30 \times 10^{-2}$ | $3.25 \times 10^{-3}$ | $2.97 \times 10^{-3}$ | $2.14 \times 10^{-3}$ | $1.34 \times 10^{-3}$ | $1.93 \times 10^{-4}$ |
| $260 \pm 3$, $-30 \pm 3$ | $7.91 \times 10^{-2}$ | $2.31 \times 10^{-2}$ | $7.96 \times 10^{-3}$ | $5.75 \times 10^{-3}$ | $3.47 \times 10^{-3}$ | $2.57 \times 10^{-3}$ | $4.97 \times 10^{-5}$ |
| $346 \pm 4$, $-10 \pm 1$ | $2.24 \times 10^{-2}$ | $8.98 \times 10^{-3}$ | $3.03 \times 10^{-3}$ | $1.86 \times 10^{-3}$ | $1.84 \times 10^{-3}$ | $1.05 \times 10^{-3}$ | $5.20 \times 10^{-4}$ |
| $346 \pm 4$, $-34 \pm 3$ | $4.58 \times 10^{-2}$ | $1.40 \times 10^{-2}$ | $5.39 \times 10^{-3}$ | $3.91 \times 10^{-3}$ | $2.90 \times 10^{-3}$ | $2.23 \times 10^{-3}$ | $3.60 \times 10^{-5}$ |
| $346 \pm 4$, $-71 \pm 7$ | $9.17 \times 10^{-2}$ | $3.25 \times 10^{-2}$ | $1.70 \times 10^{-2}$ | $7.33 \times 10^{-3}$ | $5.51 \times 10^{-3}$ | $2.88 \times 10^{-3}$ | $2.21 \times 10^{-5}$ |
| $432 \pm 5$, $-22 \pm 2$ | $3.74 \times 10^{-2}$ | $1.49 \times 10^{-2}$ | $3.22 \times 10^{-3}$ | $2.49 \times 10^{-3}$ | $1.85 \times 10^{-3}$ | $2.44 \times 10^{-3}$ | $1.25 \times 10^{-4}$ |
| $432 \pm 5$, $-52 \pm 5$ | $7.62 \times 10^{-2}$ | $4.13 \times 10^{-2}$ | $1.13 \times 10^{-2}$ | $3.23 \times 10^{-3}$ | $3.23 \times 10^{-3}$ | $4.17 \times 10^{-3}$ | $1.06 \times 10^{-4}$ |
| $432 \pm 5$, $-90 \pm 9$ | $1.77 \times 10^{-1}$ | $3.55 \times 10^{-2}$ | $1.83 \times 10^{-2}$ | $6.90 \times 10^{-3}$ | $4.75 \times 10^{-3}$ | $4.56 \times 10^{-3}$ | $2.43 \times 10^{-5}$ |

Table 5 Key features of the In-CASE setup

| Feature | Numerical value |
|---|---|
| **Collision chamber's parameters** | |
| Height of the collision chamber ($H_{In-CASE}$) | 1 m |
| Distance between droplet injection and AP injection | ≈10 cm |
| Diameter of the collision chamber | 5 cm |
| Impaction cup diameter | 2.5 cm |
| AP flowrate in the DMA | Between 0.4 and 1.2 l/min (following the selected AP radius) |
| Clean air adding at the inlet of the aerosol charger | Between 0.3 and 1.1 l/min (following the selected AP radius) |
| AP flowrate in the aerosol charger | 1.5 l/min |

| | |
|---|---|
| AP flowrate in the collision chamber ($Q_{In-CASE,c}$) | 1.5 l/min |
| Flow velocity in the collision chamber ($V_Q$) | 1.3 cm/s |
| Flowrate of the upward Argon at the inlet of AP/droplet separator | 0.4 l/min |
| Flowrate of the upward Argon in the impaction cup | 1.4 cm/s |
| AP and Argon flowrate at the outlet of In-CASE chamber (toward the HEPA filter) | 1.9 l/min |
| Air pressure in the collision chamber ($P_{air}$) | 1 atm |
| Temperature in the collision chamber ($T_{air}$) | $1.08 \pm 0.12\,°C$ |
| Relative humidity in the collision chamber ($RH$) | $95.1 \pm 0.2$ % |
| Duration of experiments ($\Delta t$) | From 3 to 6 hours (related to the expected APs mass in droplets) |
| **AP parameters** | |
| Selected dry AP radius during experiment ($a$) | 100, 150, 200 or 250 nm |
| Growth factor of the APs ($GroF$) | Between 1.73 and 1.75 (following the mean levels of relative humidity for the 4 separated AP radii) |
| Density of sodium florescein ($\rho_{fluorescein}$) | 1580 kg.m$^{-3}$ |
| Density of the wet APs ($\rho_{AP}$) | ≈1110 kg.m$^{-3}$ |
| AP terminal velocity | ≤10$^{-3}$ cm/s (equal to 8x10$^{-4}$ cm/s for the larger selected dry AP radius 250 nm) |
| AP residence time in the collision chamber | ≈80 s |
| Total AP concentration (single and multiple charged at the DMA's outlet) | From 3.10$^4$ cm$^{-3}$ (for $a$=100 nm and $q = -10 \pm 1$ $|e|$) to 2.10$^3$ cm$^{-3}$ (for $a$=250 nm and $q = -90 \pm 9$ $|e|$) |
| AP charge ($q$) | From $-10 \pm 1$ to $-90 \pm 9$ elementary charges (following the selected AP radius) |
| **Droplet parameters** | |
| Droplet radius ($A$) | $48.5 \pm 1.1$ μm |
| Droplet generation frequency ($F_d$) | 25 Hz |
| Droplet terminal velocity ($U_{A,\infty}$) | ≈25 cm/s |
| Number of injected droplets during experiments ($N_d$) | From 270,000 to 540,000 (related to the expected APs mass in droplets) |
| Observed distance between two successive droplets | ≈ 9 mm ≈ 180 droplet radii |
| Droplet residence time in the collision chamber | ≈4 s |
| Droplet charge ($Q$) | From $-3.0 \times 10^4 \pm 1.4 \times 10^3$ to $9.6 \times 10^4 \pm 4.3 \times 10^3$ elementary charges |
| Droplet charge after neutralisation ($Q$) | $0 \pm 600$ elementary charges |

| Droplet evaporation between the injection and the end of the collision chamber | ≈0.3 % |
|---|---|
| Sodium chloride concentration in the pure water | 3.3 g/l |

### 3.2.1    Effect of the product of the droplet and AP charges on the collection efficiency

The CE measurements for a wet AP radius of 432 nm are presented in Figure 7 as a function of the product of the droplet ($Q$) and AP ($q$) charges. The measurements are compared to the Dépée et al. (2019) extended model (solid line) for the 4 AP charges, considering the AP and droplet charge uncertainties. There is a good agreement between model and measurements which indicates that the analytical expression of the electrostatic forces (equation (10)) reliably describes the observations.

Indeed, an important charge influence is measured, increasing or decreasing the CE up to two orders of magnitude for large negative or positive charge products, respectively, compared to the theoretical CE value disregarding the electrostatic effects (dashed line in Figure 7). This is due to the Coulomb inverse square term in the electrostatic forces' equation (10) which dominates - attracting or repelling the APs from the droplet depending on whether the AP and droplet charges have unlike or like signs.

For small positive charge products (approximately $0 \leq q \times Q \leq 10^6 \ |e| \times |e|$), an increase of CE with a factor of more than three is measured compared to the theoretical CE value without electrostatic forces. This fact truly emphasises the contribution of the short-range attractive term in equation (10) which attracts the APs toward the droplet even though the droplet and AP charges have like signs. Indeed, as previously stated, this term prevails for small charge products (Dépée et al., 2019).

Note that the same influence of the charge product on the CE is observed for the other three wet AP radii – the CE varies up to four orders of magnitude.

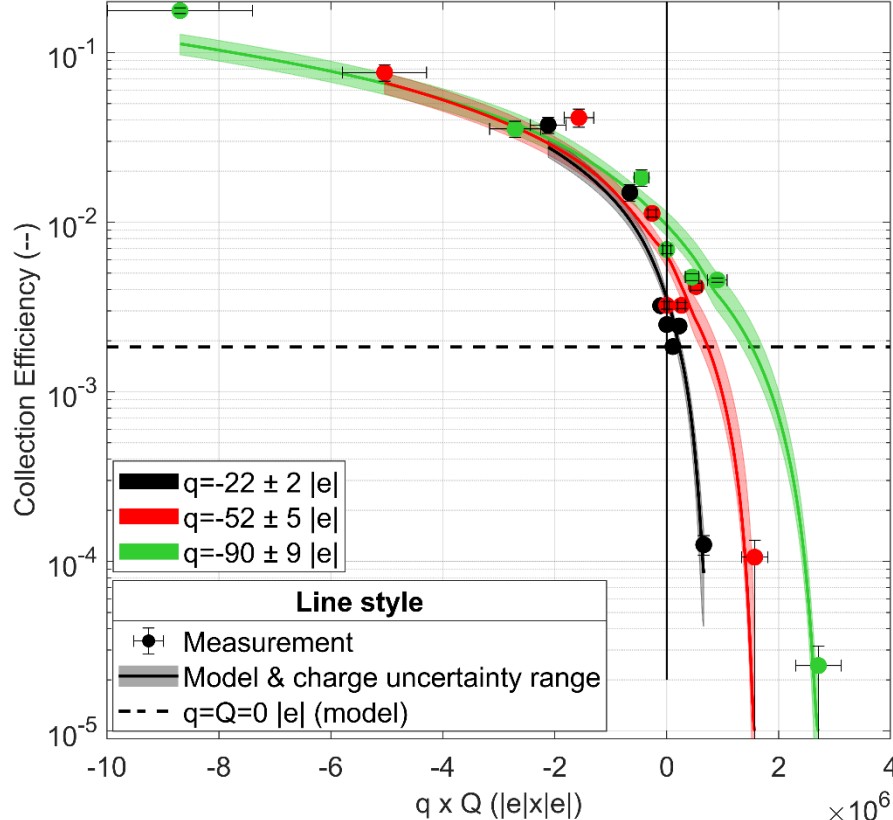

Figure 7 CE measurement as a function of the product of the droplet ($Q$) and AP ($q$) charges for the wet AP radius of 432 nm and a droplet radius of 48.5 ± 1.1 μm. The experimental conditions are summarised in Table 5. Color code informs about the AP charge. The dashed line represents the

theoretical CE value disregarding the electrostatic forces (given the air parameters 1°C, 1 atm, 95%
of relative humidity). The solid line is the interpolation of the Dépée et al. (2019) extended model
(with the charge uncertainty range) for the respective CE measurements at a given AP charge.
3.2.2    Effect of the AP charge on the collection efficiency for a neutral droplet
In Figure 8, the CE measurements (circle) for a neutral droplet ($Q = 0 \pm 600\ |e|$) are presented for
the 4 wet AP radii - referred by the color code - with the respective theoretical CE values (triangle).
The dashed line represents the theoretical CE value without electrostatic forces.
Note that the contribution of the electrostatic forces seems insignificant for an AP charge of about
-10 elementary charges and an AP radius of 346 nm and 260 nm as well as an AP of 432 nm with -20
elementary charges. Indeed, these measurements are very close to the theoretical ones with no
consideration of electrostatic forces. Several microphysical effects have probably an equivalent
contribution on the CE measurements such as electrostatic, thermophoretic and diffusiophoretic
forces, in addition to AP diffusion, weight and inertia.
However, at a given AP radius, an increase of the CE is observed when the number of elementary
charges on the APs is larger. Note that this increase appears even though the droplet is neutral (or
poorly charged considering the charge uncertainty of 600 elementary charges). For example, given
an AP radius of 346 nm, the CE is multiplied by almost a factor 4 when the AP charge increases from
-10 to -71 elementary charges. It highlights the contribution of the short-range attractive term in
equation (10), showing the presence of a surface charge distribution on the droplet formed by the
partial influence of the AP electrostatic field on it. In the current case, this is the only contribution
since the droplet is neutral and the Coulomb inverse square term is zero in equation (10). This is an
important result since, to our knowledge, there is no experimental observation of the short-range
attractive term on the CE in the previous studies of the literature. Here, the good agreement between
measured (circle) and modelled (triangle) CEs confirms that the analytical expression of the short-
range attractive term in equation (10) is reliable.
For a given AP charge, an increase of the CE is measured when the AP radius decreases, probably due
to the increase of the electrical mobility of APs. This is in line with the numerical results of Dépée
et al. (2019) even though electrostatic effects are not the only contribution involved in this CE
increase. Indeed, the Brownian motion of the APs increases for smaller APs and enhances the collision
between the droplet and APs.
Moreover, the curve slope could be increased for a decrease of the AP radius since the electrical
mobility increases but this trend is not visible in Figure 8. It can be due to the uncertainties on the
CE measurements, the droplet neutralisation and the AP charge.

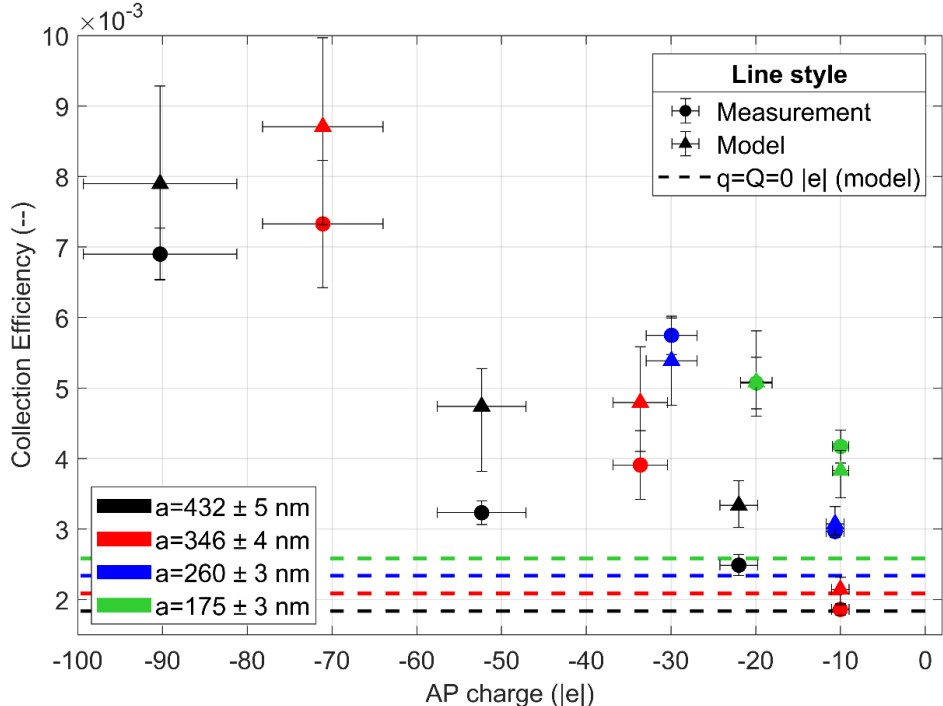

Figure 8 CE measurement (circle) as a function of the AP charge ($q$) for the 4 wet AP radii (Color
code). The respective modelled CEs are also presented (triangle). The droplet is neutral with a radius
of 48.5 ± 1.1 μm. The experimental conditions are summarised in Table 5. The dashed line represents
the theoretical CE value disregarding the electrostatic forces (given the air parameters 1°C, 1 atm,
95% of relative humidity). The vertical error bars for the modelled CEs consider the AP and droplet
charges uncertainties.
3.2.3    Comparison with existing models

3.2.3.1 Kraemer and Johnstone (1955) prediction
To describe the impact of the electric charge on the CE, there is the well-known prediction of
Kraemer and Johnstone (1955), summarised in equation (11):

$$EC_{KJ} = \left(\frac{Q}{\pi \varepsilon_0 A^2 \Delta U}\right)\left(\frac{q \times C_u}{6\pi \eta_{air} a}\right)$$

(11)

With $\eta_{air}$ - the dynamic viscosity of air, $C_u$ - is the Stokes-Cunningham slip correction factor (defined
in Appendix A of Dépée et al. (2019)) and $\Delta U$ the droplet fall velocity relative to the AP fall velocity.
This latter is assumed to be equal to $|U_{A,\infty} - U_{a,\infty}|$ where $U_{a,\infty}$ is the AP settling velocity.

Since this prediction models the contribution of the attractive Coulomb forces on the CE, only the CE
measurements with a negative charge product for the 4 AP radii are compared. In Figure 9, the
modelled CE from the prediction of Kraemer and Johnstone (1955) as a function of the measured CE
is presented. The horizontal error bars are the measurement uncertainties while the vertical ones
are the extreme theoretical CE values considering the extreme droplet and AP charges (by adding or
subtracting the charge uncertainties). It is shown that the prediction of Kramer and Johnstone (1955)
accurately describes the observation for the large charge products (red color) but the discrepancies
between model and measurement increase when the charge product decreases. Indeed, the less AP
and droplet are electrically charged, the more the model underestimates the CE compared to the
observations. This is due to the formula which only models the attractive Coulomb forces and
disregards the other effects like the AP weight, the AP inertia and the AP diffusion which tend to
increase the CE as well as the diffusiophoretic and the thermophoretic forces (**Dépée et al., 2020**).
Consequently, the prediction gives better agreement for large charge products where the attractive
Coulomb forces dominate the other effects on the AP collection. This case illustrates the strong
interest of using Lagrangian models like the one of Dépée et al. (2019) which considers all
microphysical effects involved in the in-cloud AP collection and especially their coupling.

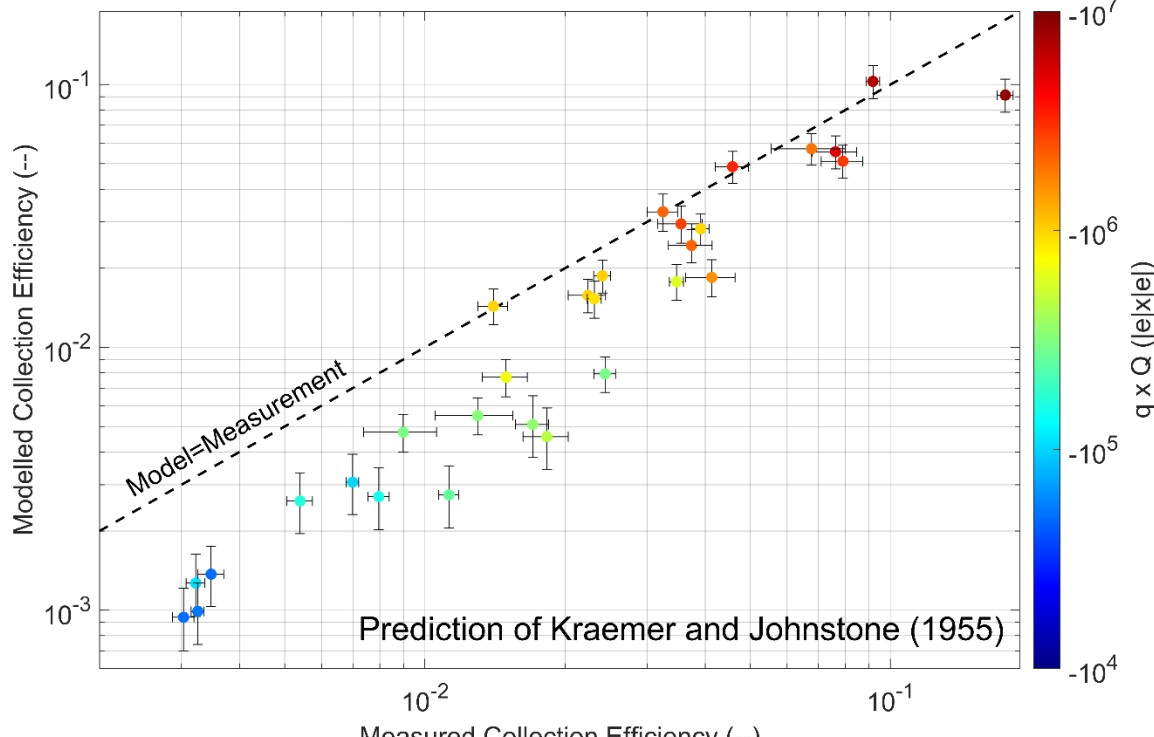

Figure 9 Modelled CE from the prediction of Kraemer and Johnstone (1955) as a function of the measured CE. The droplet radius is 48.5 ± 1.1 µm. Only the negative charge products for the 4 AP radii are considered here, represented by the color code. The experimental conditions are summarised in Table 5.

Note that Wang et al. (1983) also compared their CE measurements with this prediction, finding good agreement since they considered between -10[7] and -10[8] elementary charges on droplets and between 1 and 13.5 elementary charges on APs. So, their charge products were larger than the ones used in the present study and they had no combined effect - the attractive Coulomb force was by far the only significant contribution. It can explain why their comparison with the prediction of Kraemer and Johnstone (1955) are better than the ones presented in Figure 9. Since most of the measurements of Wang et al. (1983) are for a droplet radius of 250 µm, they are not comparable to the present ones which are significantly much smaller ($A$= 48.5 ± 1.1 µm).

3.2.3.2 Dépée et al. (2019) extended model
In Figure 10, the modelled CE from the extended model of Dépée et al. (2019) as a function of the measured CE is presented. The modelled CE are calculated from the experimental parameters (AP density, air temperature, pressure and relative humidity as well as the sizes and charges) and the CE values less than $10^{-5}$ are set to $10^{-5}$ to avoid an excessive computation time (Dépée et al., 2019). The horizontal error bars are the measurement uncertainties while the vertical ones are the extreme theoretical CE values considering the extreme droplet and AP charges (by adding or subtracting the charge uncertainties). The color code corresponds to the different droplet radii studied.
A good accordance between the model and the CE measurements are shown. Indeed, it appears that there are as many data points above the "Model=Measurement" line as below, meaning that the model overestimates as much as underestimates the observations. Thus, it can be assumed that there is no missing or unnecessary microphysics effects in the CE modelling. Moreover, the mean difference between the modelled CEs and the 70 measured CEs is 66 %. This is a reasonable value for a microphysics parameter such as the collection efficiency which varies on several orders of magnitude, especially since the value was calculated disregarding the different uncertainties (error bars in Figure 10) and was as a result over-evaluated.
Nevertheless, 6 data points seem inconsistent with discrepancies between model and measurements from 150 to 1000 %, occurring for the smallest CE values in Figure 10 (lower left). Note that the discrepancies should be even worse since the modelled CEs, set to $10^{-5}$, are actually much lower. By examining these data points, it appears that the measured AP masses in the droplet impaction cup - $m_{AP,d}$ in equation (1) - are very close to the detection limit of the spectrometer used. Moreover, for

the experimental conditions, the model predicts AP masses in the droplets lower than the detection
limit since the Coulomb inverse square term in equation (10) was very repulsive. So, the assumption
can be made that a pollution occurred during the various steps of the protocol (end of experiment,
disassembly of the chamber's bottom to reach the droplet impaction cup, change of room for the
analysis, etc.). Note that the detection limit of the spectrometer is $10^{-15}$ kg (for the nominal analysis
volume considered), which only represents ten APs with a dry radius of 250 nm deposited on the
droplet impaction cup. Thus, it exists an important uncertainty in these CE measurements related to
a possible contamination. This is difficult to quantify but the low uncertainties of the CE
measurements below $10^{-4}$ were increased in Figure 10. To reduce this potential pollution, it would
be necessary to work in a cleanroom or increase the experiment duration to avoid detection problem.
However, for these data points the experiment duration was almost 6 hours and, beyond this
duration, stability problems of the piezoelectric droplet generator were frequent.

However, a reasonable agreement between the extended model of Dépée et al. (2019) and the CE
measurements are observed. As a reminder, the mean discrepancy was over-evaluated at 66 % which
is suitable to describe a microphysical parameter varying on several orders of magnitude for the
collection efficiency. Furthermore, if the 6 inconsistent values are removed - the mean discrepancy
on the 63 remaining CE measurements decreases from 66 to 38 %.

The 38 % of discrepancy between the Dépée et al. (2019) extended model and the measurements can
be attributed to the dispersion of the AP charge distribution. Indeed, it was not possible to
characterize the AP charge distribution which remains an important uncertainty. Moreover, the AP
size distribution was assumed to be monodispersed but a dispersion exists, even if very small, which
depends on the spectral bandwidth of the DMA. This one can induce some larger (or smaller) APs
inside the AP charger which can get an electric charge significantly larger (or smaller) than the
predicted one since the charging process is roughly proportional to the AP surface. Then, in the In-
CASE chamber, some larger (or smaller) APs with a larger (or smaller) electric charge can interact
with the droplets and notably change the final AP mass collected by the droplets during an
experiment ($m_{AP,d}$). Another possible explanation is the differences in temperature and relative
humidity between the top and the bottom chamber, respectively less than 1°C and 4 % (addressed in
**Dépée et al. (2020)**). It could induce local discrepancies during the AP travel time in the chamber in
terms of AP density and radius (through the hygroscopicity) or thermophoretic and diffusiophoretic
forces which can change the likelihood of being collected by the droplets and then slightly change
$m_{AP,d}$. See **Dépée et al. (2020)** for a discussion of the influence of these two latter forces on the CE.

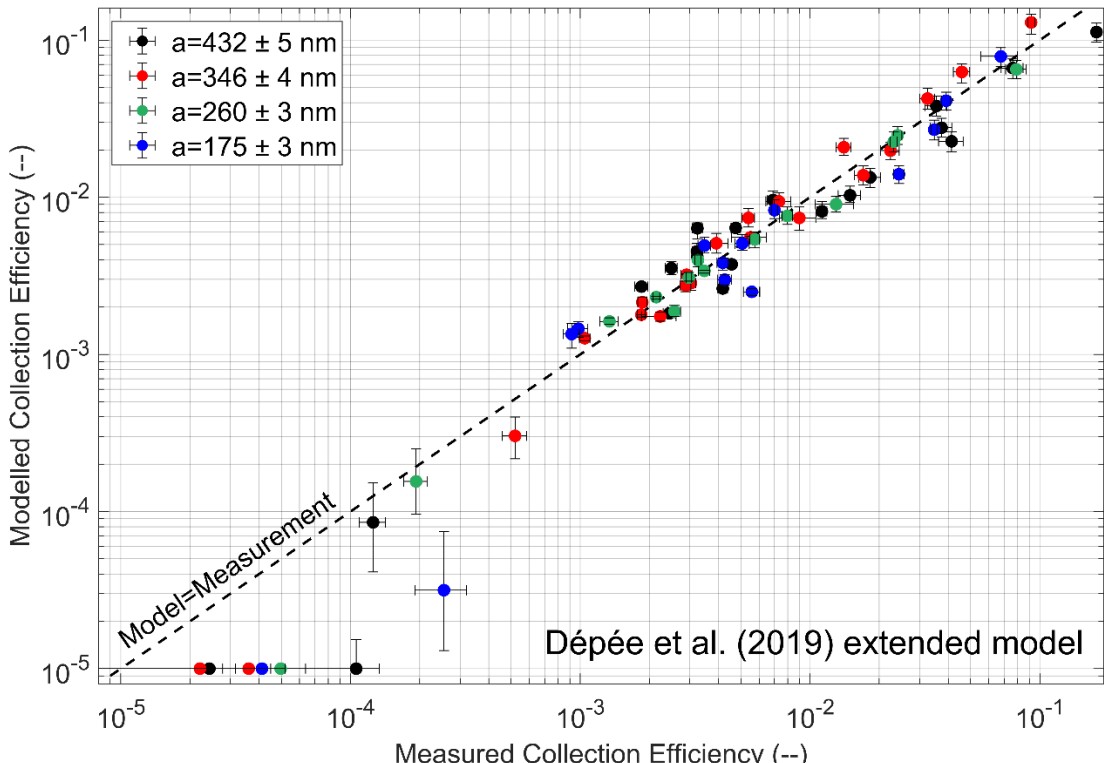

Figure 10 Modelled CE from the extended model of Dépée et al. (2019) as a function of the measured
CE. The droplet radius is 48.5 ± 1.1 μm. The color code referrers to the AP radius. The experimental
conditions are summarised in Table 5.

## CONCLUSION


In-CASE (In-Cloud Aerosol Scavenging Experiment) was developed to conduct a series of experiments
evaluating the contribution of microphysical effects on the AP collection by falling cloud droplets.
For this purpose, all the parameters influencing the collection efficiency (CE) are controlled - i.e.
the AP and droplet sizes, the AP and droplet electric charges and the relative humidity. A first
campaign was performed to study the influence of the relative humidity which is the topic of the
Part I (**Dépée et al., 2020**). This current study was dedicated to a second topic - aiming the impact
of the electric charge on the CE. Furthermore, the CE measurements allow to validate existing models
like the Lagrangian one of Dépée et al. (2019) which considers all microphysical effects involved in
the AP collection by cloud droplets. Indeed, the literature lacks baseline data to get a suitable
comparison with the modelling since most of the previous studies failed to control all parameters
influencing the CE like the AP and droplet sizes and charges as well as the relative humidity (Beard,
1974; Lai et al., 1978; Barlow and Latham, 1983; Byrne and Jennings, 1993). Even though some studies
stand out (Wang and Pruppacher, 1977; Wang et al., 1983), no one examined the influence of the
electrostatic forces when the droplet and AP charges had like signs. Thus, the short-range attractive
term from the analytical expression of the electrostatic forces - equation (10) - used in the current
Lagrangian models (Tinsley and Zhou, 2015; Dépée et al., 2019) has never been experimentally
validated or at least emphasised.
In the new CE dataset, the APs and droplets are accurately charged through custom-made droplet
and AP chargers detailed above. Since both charge polarities are found in clouds (Takahashi, 1973),
the droplets were negatively as well as positively charged during experiments. Moreover, several
amounts of elementary charges on the droplet were considered to represent a neutral droplet but
also the weakly and strongly charged droplets respectively found in stratiform and convective clouds
(Takahashi, 1973). The AP charge varied from zero to -90 ± 9 elementary charges depending on the
AP size to represent different amounts of elementary charges encountered in the atmosphere,
particularly the ones of radioactive APs. The relative humidity was maximised in this experimental
work (**95.1** ± 0.2 %) with a mean temperature in the collision chamber (1.08 ± 0.12 ˚**C**) stable and
comparable with the other study in the companion paper: Part I (**Dépée et al., 2020**). Thus, the

thermophoretic and diffusiophoretic contributions on the CE measurements were reduced as much as possible. Nevertheless, since Dépée et al. (2020) measured a significant contribution for a comparable relative humidity level, these two forces were still added to the Dépée et al. (2019) model for a reliable model/measurement comparison. Finally, the droplet radius was 48.5 ± 1.1 μm and 4 wet AP radii were used - from 175 ± 3 to 432 ± 5 nm. Note that the hygroscopicity of the sodium fluorescein salt was considered in the calculation of the wet AP radius and the AP density.

From the 70 measurements obtained, an influence of the electric charges of 4 orders of magnitude on the CE was observed, strongly increasing or decreasing the CE for large charge products, respectively negative or positive. An increase of the CE was also measured by considering more elementary charges on the APs even though the droplets were neutral (within uncertainties). This observation is an important result since it emphasises the contribution of the short-range attractive term in the electrostatic forces (equation (10)). It validates a surface charge distribution on the droplet, formed by the partial influence of the AP electric field on it, which has never been experimentally shown, to our knowledge, in the literature before.

The CE measurements with opposite signs on the droplet and AP were compared to the prediction of Kraemer and Johnstone (1955), giving good agreements for large negative charge products where the Coulomb attractive forces prevail over the other dynamic effects. This is in line with the work of Wang et al. (1983) who also obtained good agreement, considering another droplet radius (250 μm) and larger negative charge products. However, in the present study, an increase of the discrepancies between the prediction and the measurement was measured when reducing the number of elementary charges. This is due to the electrostatic forces not being the only effect involved in the AP collection. There is actually a coupling of electrostatic, diffusiophoretic and thermophoretic forces as well as the AP diffusion, weight and inertia. Thus, when the charge product is not strong enough (gets significantly smaller than $10^7$ $|e|\times|e|$), Lagrangian models as the one of Dépée et al. (2019) remain the best estimation of the CE.

Finally, the CE measurements were also compared to the extended model of Dépée et al. (2019), showing a really good description of the observed effects. Indeed, the mean discrepancy of the modelling and the 70 measurements was 66 % which is suitable for a microphysical effect varying on several orders of magnitude like the collection efficiency. This value was even better when 6 inconsistent measurements, probably contaminated, were disregarded – as it decreases from 66 % to 38 %. Moreover, note that the model overestimates as much as underestimates the observations so that the discrepancies probably result from remaining uncertainties (like the dispersion of the AP charge distribution) instead of a missing microphysical effect in the CE modelling.

To conclude, 70 new CE measurements are now available that include the influence of the electric charges, showing significant differences with the previous CE measurements and theoretical values from the literature which disregard the electrostatic forces. Thus, it appears to be essential to study the impact of the new baseline data in a cloud-model like DESCAM (Detailed SCAvenging Model, Flossmann, 1985) to examine the influence of the electric charges on the total wet AP removal in the atmosphere. It could strongly affect the atmospheric AP removal since cloud droplets are known to be charged (Takahashi, 1973) as well as the atmospheric AP, even more when APs are radioactive. Indeed, Dépée et al. (2019) estimated that the electric charge of the radioactive APs emitted after the Fukushima accident in 2011 was up to 600 elementary charges. Thus, AP removal could be substantially affected by the electrostatic forces in-cloud and significantly change the ground contamination after a discharge of radioactive materials from a nuclear accident. Since the new Lagrangian model of Dépée et al. (2019) showed an accurate description of the influence of the electric charges (and also of the relative humidity, studied in Part I **(Dépée et al. (2020)**) on the CE, this latter constitutes a simple, convenient and rapid manner to obtain a CE evaluation for its incorporation in cloud models.

# Appendix A - AP charger

## A.1 AP charging relationship's acquisition

The AP charging relationships were obtained by performing *ex situ* experiments with the setup presented in Figure A.1. A nominal AP flow goes through the charger with a monodispersed AP size distribution. At the charger's outlet, the flow of charged AP is subdivided - 0.6 l/min is directed to a Condensation Particle Counter (CPC; TSI 3787) to deduce the concentration number of AP in the charger ($C_{N,AP}$) while the other part goes toward an electrometer (TSI 3068A) to measure the current ($I_{elect}$) due to the charge evacuation. Before entering the CPC, APs are neutralised to avoid any deposition on the metallic walls of the CPC and then the AP flow passes through a diffusion battery to filter the fine particles produced during the discharges inside the charger. The mean AP charge ($\langle q \rangle$) was then calculated from the equation (A.1) with the elementary charge ($e$) and the AP flowrate in the electrometer ($Q_{elect}$):

$$\langle q \rangle = \frac{I_{elect}}{e \times C_{N,AP} \times Q_{elect}} \tag{A.1}$$

Several AP flowrates in the charger ($Q_{charger}$) were considered to study the AP penetration. When $Q_{charger}$ was less than 0.7 l/min, clean air was added before the CPC to maintain a CPC flowrate of 0.6 l/min - this part is presented in red in Figure A.1. From these experiments, it was found that $Q_{charger}$=1.5 l/min maximises the AP penetration through the charger. Note that the AP penetration is defined, at the charger's outlet, as the AP number concentration when the charger is switched on over the AP number concentration when this latter is switched off.

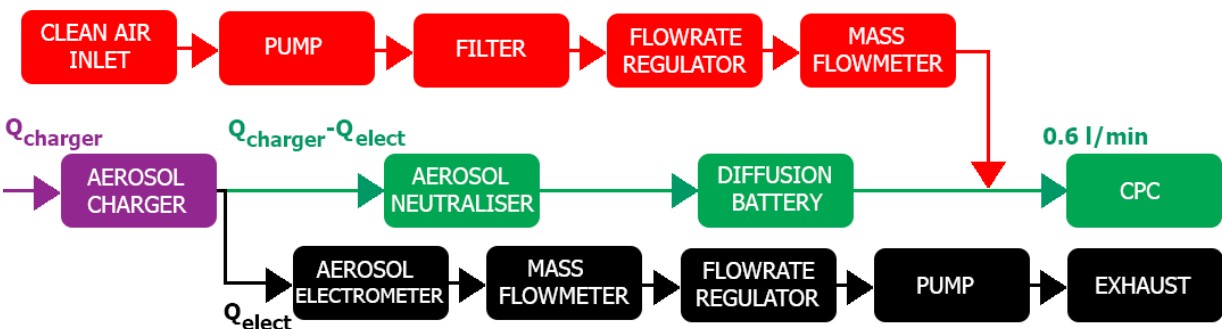

Figure A.1 Setup to obtain the AP charging relationship. The red dashed rectangle is the part added to study the AP penetration through the charger.

## A.2 Validation

The setup (Figure A.1) was conducted with the charger turned off to measure the mean AP charge after the DMA. It was found one elementary charge on APs which validates that the multiple charged APs are stopped at the DMA's inlet by the aerodynamic impactor. Thus, the assumption made that the AP size distribution is monodispersed after the DMA is justified. The AP charge was also analysed during 5 hours - no AP charge modification was measured over time. Moreover, the saturated AP charges visible in Figure 4 for a tungsten wire potential less than -12.5 kV was also compared to the theoretical values of Pauthenier and Moreau-Hanot (1932) - giving a good agreement.

Note that two other characterizations were made during these *ex situ* experiments like the determination of the ion current between the grounded cylinder and the tungsten wire (Figure 3) or the discharge frequencies - these both parameters are related to the tungsten wire potential. These curves were used to precisely identify the discharge regime of the charger (Unger, 2001) - the negative Trichel regime which provides a large discharge frequency and then a spatially homogeneous particle charging around the tungsten wires.

# Appendix B - Droplet charging relationship obtention

## B.1    Overview

Figure B.1 presents the setup used in *ex situ* experiments to measure the droplet charge where the charging relationship in Figure 6 comes from. The housing made with a 3D printer - containing the droplet injector and the charging system (detailed in section 1.5) - is set above a capacitor composed of one neutral potential plate and another plate connected to a high potential ($V_{cap}$). In this latter, pictures are obtained by optical shadowgraphy to get the droplet trajectories. The electric field ($E_{cap}$) induced in the capacitor disturbs the droplet motion according to its electric charge. Thus, the droplet charge is evaluated by finding the one which fits the best the theoretical droplet trajectory - deduced from the 2$^{nd}$ Newton's law - and the measured droplet trajectories. A Faraday cage ensures the electric field at the capacitor ($E_{cap}$) has no effect on the electric field at the electrostatic inductor ($E_{ind}$). Since this is not a proper Faraday cage because of the holes for droplets and camera, a horizontal metallic perforated plate is added below the droplet generator housing and connected to the neutral potential to prevent the electric field ($E_{cap}$) from changing the droplet charge.

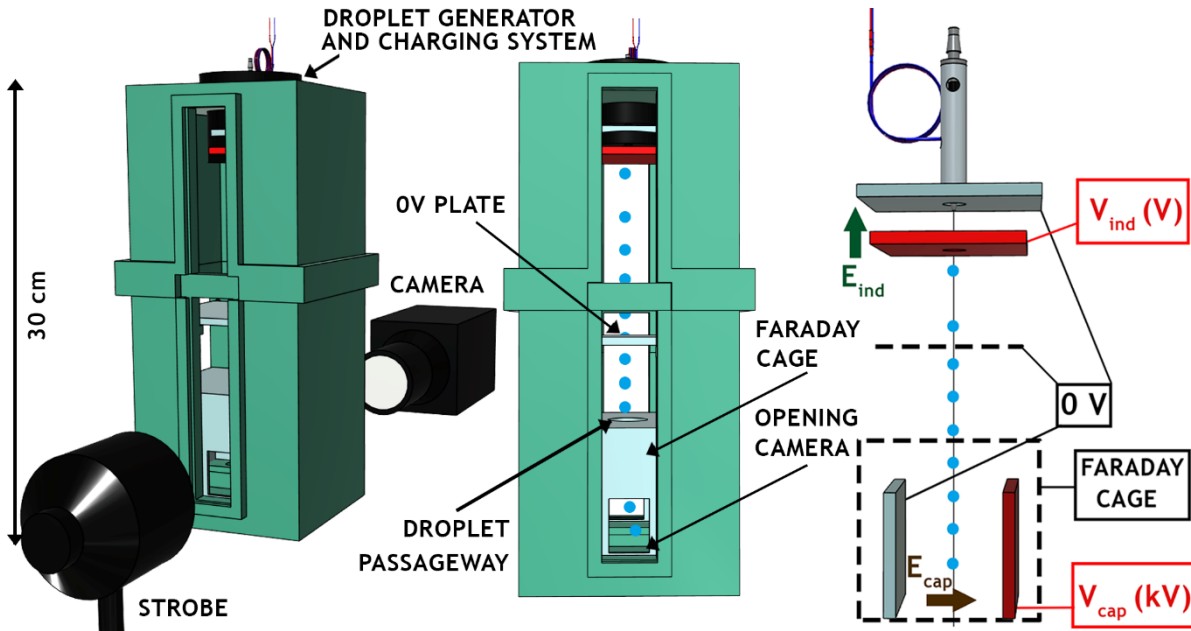

Figure B.1 Setup to obtain the droplet charging relationship - (**Left** and **Center**) 3D view and (**Right**) schema (not at scale)

## B.2    Droplet charge evaluation

A series of 200 pictures pairs, with each one dephased from the other by a known time-step ($\Delta t$), are obtained by optical shadowgraphy at the level of the capacitor. A circle Hough transform is then applied to evaluate the droplet centers in every picture - an example is given in Figure B.2 (Left) where the two droplets from a picture are represented by two black crosses meanwhile the blue cross is the detected droplet from the coupled picture dephased by $\Delta t$.

Then, the instantaneous droplet velocity $\overrightarrow{U_{D_0}(t)} = U_{D_0,x}\,\overrightarrow{u_x} + U_{\infty,A}\,\overrightarrow{u_y}$ at the first detected droplet position ($D_0$) of coordinates ($x_{D_0}, y_{D_0}$) is calculated and the vertical velocity component ($U_{\infty,A}$) determines the droplet radius ($A$) by reversing the Beard (1976) model. Here, the circle Hough transform is not used to calculate the droplet radius like during CE experiments - see Figure 8, right from part I, Dépée et al. (2020). Indeed, the camera zoom is at the lowest to get a large field - the uncertainty would be too large.

Finally, the theoretical droplet trajectories at the capacitor are deduced by solving the 2$^{nd}$ Newton's law where the buoyancy force ($F_{buoy}$), the drag force ($F_{Drag}$) and the electrostatic force ($F_{E_{cap}}$) related to the electric field ($E_{cap}$) at the capacitor are considered, summarised in equations (B.1) :

$$\begin{cases} m_D \dfrac{d\boldsymbol{U}_D(t)}{dt} = \boldsymbol{F}_{buoy} + \boldsymbol{F}_{Drag} + \boldsymbol{F}_{E_{cap}} \\[2mm] \boldsymbol{F}_{buoy} = -m_D \dfrac{\rho_{water} - \rho_{air}}{\rho_{water}} g\, \boldsymbol{u}_y \\[2mm] \boldsymbol{F}_{Drag} = -\dfrac{C_D \pi \rho_{air} U_D{}^2 A^2}{2} \dfrac{\boldsymbol{U}_D(t)}{\|\boldsymbol{U}_D(t)\|} \\[2mm] \boldsymbol{F}_{E_{cap}} = Q E_{cap} \boldsymbol{u}_x \end{cases} \tag{B.1}$$

With $\boldsymbol{U}_D$ - the instantaneous droplet velocity vector at the computational time $t$, $\rho_{air}$ and $\rho_{water}$ -
the air and water densities, $g$ - the acceleration of gravity, $m_D$ - the droplet mass, $Q$ - the droplet
charge, $C_D$ - the drag coefficient, $\boldsymbol{u}_x$ and $\boldsymbol{u}_y$ - the unit vectors in the cartesian coordinate system
visible in Figure B.2 (Left).
By projecting on the corresponding axis, it is obtained the system of equations (B.2) to solve:

$$\begin{cases} m_D \dfrac{dU_{D,x}(t)}{dt} = Q E_{cap} - \dfrac{C_{D,x} \pi \rho_{air} U_{D,x}{}^2 A^2}{2} \\[2mm] m_D \dfrac{dU_{D,y}(t)}{dt} = -m_D \dfrac{\rho_{water} - \rho_{air}}{\rho_{water}} g - \dfrac{C_{D,y} \pi \rho_{air} U_{D,y}{}^2 A^2}{2} \end{cases} \tag{B.2}$$

Where $C_{D,x}$ and $C_{D,y}$ are the drag coefficient projections depending on the Reynolds number
projections $Re_x$ et $Re_y$ in the cartesian coordinate system. Since $Re_x \ll 1$ et $Re_y < 2$ in the study, the
drag coefficient projections are calculated from the analytical expression given by Hinds (2012) and
summarised in equations (B.3):

$$\begin{cases} C_{D,x} = \dfrac{24}{Re_x} = \dfrac{12\,\eta_{air}}{A U_{D,x} \rho_{air}} \\[3mm] C_{D,y} = \dfrac{24}{Re_y} \underbrace{\left(1 + 0{,}15 Re_y{}^{0,687}\right)}_{①} \approx \dfrac{12\,\eta_{air}}{A U_{D,y} \rho_{air}} \underbrace{\left(1 + 0{,}15 \left(\dfrac{2 A U_{\infty,A} \rho_{air}}{\eta_{air}}\right)^{0,687}\right)}_{=K_1} \end{cases} \tag{B.3}$$

Note that the term ① in the Equations (B.3) is supposed as constant to simplify the resolution of the
equations (B.1) - giving second order differential equations. This assumption is justified since $Re_y$ is
close to the unity and then $C_{T,y} = \dfrac{24}{Re_y}$ remains suitable. The equation system to solve becomes,
equations (B.4):

$$\begin{cases} m_D \dfrac{dU_{D,x}(t)}{dt} = Q E_{cap} - \overbrace{6\pi A \eta_{air}}^{K_2} U_{D,x} \\[2mm] m_D \dfrac{dU_{D,y}(t)}{dt} = -m_D \dfrac{\rho_{water} - \rho_{air}}{\rho_{water}} g - \underbrace{6\pi A \eta_{air} K_1}_{K_3} U_{D,y} \end{cases} \tag{B.4}$$

After two consecutive integrations with the initial conditions - $U_{D,x}(t=0) = U_{D_0,x}$, $U_{D,y}(t=0) = U_{\infty,A}$,
$(x_D(t=0), y_D(t=0)) = (x_{D_0}, y_{D_0})$, the analytical equations of the horizontal and vertical droplet
positions, respectively referred as $x_{th}$ and $y_{th}$, are given in equations (B.5):

$$\begin{cases} x_{th}(t) = \dfrac{Q\,E_{cap}}{K_2} t + \dfrac{m_D}{K_2}\left(U_{D_0,x} - \dfrac{Q\,E_{cap}}{K_2}\right)\left[1 - e^{-\frac{K_2}{m_D}t}\right] + x_{D_0} \\[3mm] y_{th}(t) = -\dfrac{m_D(\rho_{water} - \rho_{air})}{K_3 \rho_{water}} gt + \dfrac{m_D}{K_3}\left(U_{\infty,A} + \dfrac{m_D(\rho_{water} - \rho_{air})}{K_3 \rho_{water}} g\right)\left[1 - e^{-\frac{K_3}{m_D}t}\right] + y_{D_0} \end{cases} \tag{B.5}$$

Where $E_{cap} = -grad(V) = -\dfrac{V_{cap}}{0{,}01}$ V/m.
As presented in Figure B.2 (Left), for every pair of pictures, the droplet charge ($Q$) is then evaluated
by looking for the theoretical droplet trajectory from the Equations (B.5) which fits the best with the
observed droplet positions. In the given example (Figure B.2, Left), the fitted theoretical trajectory
- for $V_{ind}$ = -32.25 V, $V_{cap}$ = -629.5 V, $A$ = 49.5 µm and the air temperature $T_{air}$ = 292.55 K - illustrated
by the red line is obtained for a droplet charge ($Q$) of +9.10e+04 |e|. Finally, this method is applied
for the 200 picture pairs to get the mean droplet charge value - visible in Figure B.2 (Right). Note
that the standard deviation of the 200 $Q$ values gives the error bars in Figure 6.

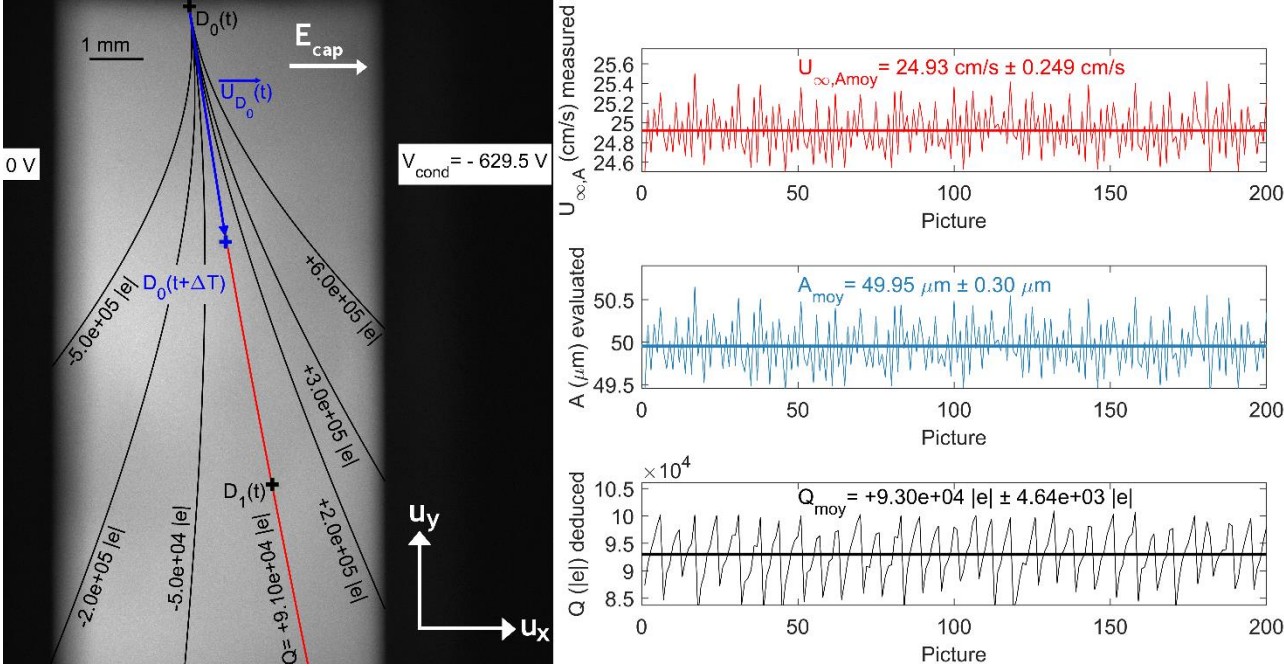

Figure B.2 (**Left**) Determination of the theoretical droplet trajectory which fits the best with the
observed droplet positions - red line - and deduction of the droplet charge ($Q$). In this example,
$V_{ind}$ = -32.25 V, $V_{cap}$ = -629.5 V, $A$ = 49.5 μm and the air temperature $T_{air}$ = 292.55 K. (**Right, Top**)
Terminal velocity measurement, (**Right, Middle**) Droplet radius evaluation by reversing the Beard
(1976) model and (**Right, Bottom**) droplet charge deduction for a series of 200 pictures pairs. Mean
and standard deviations for the corresponding parameters are presented.
**B.3    Validation**
The method presented at the previous section is possible as long as the droplet has reached its
terminal velocity. As mentioned in **Dépée et al. (2020)** and visible in Figure 8 (Left) of the same
paper, droplets are generated at a velocity larger than their terminal velocity. It has been found that
a distance between the droplet generator and the capacitor of 15 cm was large enough to allow
droplets to reach their terminal velocity. In the setup in Figure B.1, this requirement prevails.
An experiment was performed to ensure that reversing the Beard (1976) model was a suitable method
to evaluated the droplet radius. For this purpose, the same droplet train was recorded in optical
shadowgraphy with a camera zoom at the lowest and at the greatest to respectively apply the Beard
(1976) model inversion and the circle Hough transform. In all tests, it was found a discrepancy of less
than 2 % between the two methods, giving overvaluations as well as undervaluations when comparing
one to the other.
Also, the disturbance of the electric field at the capacitor ($E_{cap}$) on the vertical droplet velocity was
studied. $E_{cap}$ was then turned on and off to investigate the change in vertical droplet velocity. It was
found that during tests, $E_{cap}$ reduced the vertical velocity up to 1.3 %. This situation was for a droplet
charge ($Q$) and a capacitor potential ($V_{cap}$) both negative. Some other tests also showed that the
droplet vertical velocity was increased up to 0.3 %, for a droplet charge and a capacitor potential of
unlike sign. Since these two extreme cases respectively represent an undervaluation of less than
0.7 % and an overestimation of less than 0.2 % of the droplet radius - this effect was neglected.
Finally, two other validations can be formulated by examining Figure 6. First, several capacitor
potentials ($V_{cap}$) were used in the tests - from -629.5 to -477.4 V - giving the same charging
relationship. The Faraday Cage is consequently reliable, there is not impact of the electric field
($E_{cap}$) on the droplet charge. Secondly, in the four tests the droplet radius varies from 47.0 to
     51.2 μm. Thus, the droplet charging system is independent of the droplet size and droplet
     evaporation.

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
