# Peer review of "Laboratory study of the collection efficiency of submicron aerosol particles by cloud droplets. Part II - Influence of electric charges."

_Atmospheric Chemistry and Physics, 2020_

## Referee Comment (RC1) · Anonymous Referee #1 · 14 Sep 2020

Review for *Laboratory study of the collection efficiency of submicron aerosol particles by cloud droplets. Part II - Influence of electric charges* by Dépéé et al. The paper presents an experimental of collection efficiency of submicron aerosol particles by cloud droplets under high humidity while testing different electric charge setups. This is a very interesting work with a very nice setup. I believe the paper fits well with ACP. My only issue is that the paper needs some organization clarification and rewriting of some parts to be suitable for publication.

**General comments**

You are citing your work part 1 before it has been published which is a bit problematic. I do not recommend using this (unless the editor disagrees) at lease until it has been accepted. You have too much depended on it in this paper which makes it hard to understand can this paper stand by itself.

I had to read your other paper (part 1) to understand this paper, as you refer to it in almost every section. You need to provide more explanation (instead of sending us to part 1) and refer to reader to the part 1 only in case they would like to read something in more detail

I think a better organization of your experimental set up is required. As a reader, I had to jump back and forth to understand what you were referring to. Try to combine the different sections into your general set up description. You can also use fewer words to describe thins as thermodynamic conditions. You can write this information in a 1-2 sentence. Also, you can add Fig 3 and 5 as part of the setup figure with sections of Fig A, Fig B. Please, and tell us what exactly we see on the figure and where, as it is not understood. For example, I do not understand where is the location of your piezoelectric injector? Does the aerosol pass the humidifier area? This means you increase their size due to the hygroscopic process, have you evaluated that aspect? Also, what is your aerosol size distribution, you mention sizes in the result (not the method) but do not show it. In the method part, you do not mention if the droplets stay stable or grow in your chamber since RH is 95%, you mention it later in the result. This led me to the last part you barely mention anything on uncertainties in the method later in the result you explain some of the differences you got by explaining them.

Overall, I think you should organize the paper better it feels like you are jumping between your experimental results and the model which make it's hard to follow

**Specific comments**
**Abstract**
In general, it is not recommended to use references in the abstract. Just keep it general and mention you will compare to literature or expend other works.

**Introduction**
Lines 37-38 the use of this example *developing cardiovascular disorder (Crouse et al., 2012)* is wrong, as the impact is not just cardiovascular, I recommend to keep it general - *They are also a key topic in human health where Aps increase the likelihood impacting human health (morbidity and mortality) (new citations).*

Lines 40-41 you wrote h*ave been investigated extensively over the last decades* but you do not cite a single paper to show some of these papers.

Line 43 I suggest remove the word the (*the modeling*)

Lines 49 -50 can you change the sentence – this is very confusing - *leave the streamline that surrounds the falling droplet* maybe you can say that they are swept by the streamline of the droplets or something like that.

Line  51 change the word *motion*
Line 53 change the word *massive*
Line 54 use other words than *to go*

Lines 61-62 instead or writing *These effects prevail in a subsaturated air - as it is the case sometimes in clouds - and are discussed in Part I (Dépée et al., 2020).* I would prefer to also see a description of them here which will make the reader life easier to understand the prices

Lines 68-71 I believe your example is not good, just state that AP is charged by natural atmosphere phenomenon e.g. lightning process, dust storm. The use of this example *particularly for nuclear safety issues when the APs removal by clouds result from the discharge of radioactive materials from a nuclear accident* is so low compared to all other phenomena, I do not recommend using it. Besides, I think you should mention that electroscavenging is more important for smaller particle sizes (<0.1 μm) than Brownian diffusion as found by the model work of Tinsley et al. (2001) and Ardon-Dryer (2015).

Lines 75-90 the entire paragraph is not clear, I recommend rewriting it, also provide the eq here since it is relevant for this paper.

Lines 91-106 will be nice to organize this and write this in a way that will flow better, I provide below small examples to improve this paragraph. Organizing all information in a table could help the reader to understand similarities and differences between previous works.
Lines 91-92 just say this instead *Several laboratory studies investigated the influence of the electric charges on CE* (citations)
Line 94 change *Beard (1974) do not,* to *Beard (1974) did not*
Ling change *Lai et al. (1978) have a polydispersed*, to *used a polydispersed*

Lines 111-113 please rewrite, even try to combine it with the previous sentence.

Lines 114- 124 I think is not relevant just add the comparison in the relevant parts in the paper.

**Experimental Setup**
Explain some of the measurement uncertainties, as you will mention later in the result but you do not mention anything in the method part.
How do you know the droplets reduced in size by 0.5% is it based on calculation or measurements?
What do you mean by Penetration tests?
I think your information in section 2.1 and Table 1 should be in the setup section as it describes the particles you used for the experiment.
Please describe the main Uncertainties that may impact your results in this paper

Lines 166 -169 you do not need this, this paper stand by itself, if you insist using the reference of part 1 at least explain how the set up here is different.

Line 185-186 instead of sending us to the paper just write in short here. Here you said RH was kept at95% in the other paper as 71% so I am confused that's why you need to write it clearer here.

In Fig 6 you show different droplets sizes, how do you know these sizes how they were evaluated.
Line 241 – you cite Ardon-Dryer et al. (2015) but it is not your reference list

Lines 284-285 information of the duration of each experiment should have been provided in the experimental section
Line 285 - growth factor of what droplets or aerosol

**Results and discussions**
Line 308 - Extension of the Dépée et al. (2019) model,
Maybe this should be in your method parts and nor as part of your result
Lines 312-313, unclear what do you mean?
Lines 316-317 hard to follow what belongs to what, which Eq 9? perhaps add them as an appendix

Line 324 and Table 2- where are these 4 AP sizes come from, in the method you presented other sizes. what do you mean wet AP, you mention it here for the first time?
Consider presenting Table 2 as a plot it is hard to see any connection between the variables.
Line 367 -  what about the size or charge of the droplets was it kept constant? In the table, before you mention different charges, which one was used here
Line 346 - What do you mean have like signs
Line 349-350 – what do you mean this is not clear
Line 362-364 **–** seems like something as figure caption more than information on what presented in fig 8
Line 368 -370 this is not true, you write as if forces are equal, but we know they are not. Depending on the particle size
Lines 398-399 – rewrite the sentence add information about why it is important to do it
Line 415 no need for *(when color goes to blue).*
Lines 448-449, I disagree perhaps you have something in the model that has a bigger impact and it causes that uncertainties and disagreement with the measurements.
Lines 461-463 Unclear, these are also something you should talk about in the method part
Lines 468 – 470 – information that should be in the method also
Line 472-473 I think you should start this section with this, talk about the agreement up to $10^{-3}$ and then talk about the disagreement
Lines 478-492 you can calculate CE while changing AP sizes as suspected in the chamber their size might change, this way you could see if this can explain the differences you see

**Conclusion**
Your conclusion looks more like a discussion then a conclusion, I recommend organize it differently.

**Reference**
For forgot to put your 2019 paper
For your 2020 paper, you can at least write submitted and to where, since it is ACP it should have a DOI number

**Comments on Figures**
Figure 1.
I am not a big fan of Fig 1. It is too complex and confusing, too much information (AP and droplets size, flow charge, etc). Since you are describing, in general, the different mechanisms that impact the CE processes, I would prefer to see something more general as Fig 1. in Ardon-Dryer (2015).

Figure 2.

[Figure]

What is this part for: <small>CLEAN AIR INLET</small> and where is the piezoelectric injector it is not clear from the figure?

Figure 5 is very similar to a figure 5 you have in a paper in part 1 do you need it here.
Figure 9 what size of AP was used here?

Figure 10 - *The color code referrers to the droplet radius*, I believe you meant AP right not droplets

Appendix
I believe the numbers of the appendix should not continue you should give them new numbers as the appendix is a different part of the paper. Also, it would have been nice to see parts of the appendix as part of the paper when you explain your set up so we can understand in depth what you have done.

Reference used in this review
Ardon-Dryer, K., Huang, Y.-W., and Cziczo, D. J.: Laboratory studies of collection efficiency of sub-micrometer aerosol particles by cloud droplets on a single-droplet basis, Atmos. Chem. Phys., 15, 9159–9171, https://doi.org/10.5194/acp-15-9159-2015, 2015.

Tinsley, B. A., Rohrbaugh, R., and Hei, M.: Electroscavenging in clouds with broad droplet size distributions and weak electrification, Atmos. Res., 59, 115–135, 2001.

---

## Referee Comment (RC2) · Anonymous Referee #2 · 10 Nov 2020

**Reviewer #2 comments**
Dépée et al., *"Laboratory study of the collection efficiency of submicron aerosol particles by cloud droplets. Part II - Influence of electric charges".*

- For information, this reviewer is also a reviewer #2 of the companion paper by Dépée et al.: *"Laboratory study of the collection efficiency of submicron aerosol particles by cloud droplets. Part I - Influence of relative humidity".*

- The present paper presents a measurement of the collection efficiency of aerosol particles by sedimenting raindrops under various electrical charge states of the raindrops and aerosol particles. The electric force between a charged aerosol particle and a charged raindrop has two components: 1) the long-range Coulomb force between charges and 2) the short-range force due to an induced image charge distribution on the raindrop (independent of its net charge state) by the charged aerosol. The latter force is always attractive and dominates when the aerosol is within a few droplet radii of the droplet surface, whereas the former dominates at greater distances and is either attractive (for unlike charges) or repulsive (for like charges). These mechanisms are collectively known as electroscavenging of aerosol particles.

- Cloud droplets – even in warm clouds – are frequently charged. Moreover, aerosol particles also may become charged at the few 10e levels due to the evaporation of charged droplets or space charge effects at the edge of cloud layers in Earth's electric field. So electroscavenging is likely to be an important microphysical process for atmospheric aerosol and cloud microphysics. However, although it has been fairly extensively treated theoretically, there are very few experimental data on electroscavenging. There was considerable experimental effort on the subject in the 1970-80's, but with limited experimental control and resulting measurements that differed by several orders of magnitude.

- The experimental measurements reported in this paper are therefore essentially unique. The experiments are carefully executed. The authors have demonstrated in Part I of their laboratory study that their experimental apparatus is capable of precise measurements of the raindrop-aerosol capture efficiency (CE) (in the former case, measuring the effects of thermophoresis and diffusiophoresis). The authors have already published a theoretical model that includes electroscavenging (Dépée et al., 2019), which agrees well with a similar model previously reported by Tinsley and co-authors. As for the Part I paper, I have no hesitation to recommend that the Part II paper also be published in ACP, after responding to the comments below.

**General comments**

- I recommend that the authors apply the same general comments that I made for the Part I paper also to the present paper. In particular the authors should ask a native English speaker to edit the manuscript for poor English grammar and lengthy sentences.

- There is a lot of cross-referencing with the Part I paper, and many parts (main apparatus description, derivation of the collection efficiency, …) where texts (and some figures) are repeated almost verbatim. I leave it to the editor and authors to decide if it is best to keep these as two separate papers or to make the Part I and Part II papers as two large chapters of the same paper.

- The most surprising aspect of the paper is that the authors have made some unique and impressive measurements *and* have developed a careful theoretical model yet there is only one figure (Fig. 10) where they compare their measurements with their model. Why do they omit any comparison of their model with their data in the other two figures (Figs. 7 and 10) where they are presented? Without this comparison it is hard to get confidence that their data do indeed verify the short range image force attraction. (I will pick this point up below.)

**Specific comments**

l.27: Replace "the neutralisation" with "zero".

l.30: Replace "correlation of Kraemer and Johnstone (1955)" with "prediction of Kraemer and Johnstone (1955)" (and elsewhere in the text).

l.34: Replace "on" by "on".

l.49 Replace "is mainly depending" with "mainly depends".

l.71-74: Please clean up this sentence.

l.114-115: Please clean up this sentence.

l.121: Remove "a".

Fig.1: See the comments I made for Part I. I suggest you add some more trajectories to panel D that show aerosol particles outside the geometrical path of the raindrop, which are attracted into a collision (or not).

l.141: Indicate the AP (air) flow velocity and transfer time in the chamber.

l.144: Replace "go out" with "pass out of".

Fig. 2: You do not show in Part II but we see from Part I that the APs are introduced into the sheath region of the laminar flow down the tube, whereas the droplets fall down the centre of the tube. The APs have charges between 25e and 150e. What is their number concentration (cm-3)? I could not find this anywhere in the text and it is an important number. The APs will form a space charge in the tube that pushes them to the walls and away from the central region where the droplets fall. How big is this effect? Does it influence your estimate of the AP number concentration seen by the falling droplet?

Fig.2: You estimate the mean AP number concentration in the main tube with the HEPA filter. Charged aerosol will have higher losses in the main tube and in the pipes leading to the neutralizer. How big is this loss and is it corrected for when estimating the mean AP number concentration in the main tube?

l.175: Replace "the atmospheric one" with "one atmosphere".

l.181: Replace "So," with "In this way".

l.182: Replace "was" with "were".

l.198: replace "get" with "are".

l.210: Replace "variating" with "varying".

Fig. 5: This is a very poor figure. The "3D printing" is a black blob with no detail. It conveys no information. And what is a "3D printing"? If you mean that the piezoelectric droplet generator is installed in a housing made with a 3D printer, then state that. I suggest a simple line schematic cross section should be provided to replace the three objects in this figure.

Fig. 6: Replace "Charging low of the electrostatic inductor colors" with "Droplet charge versus electrostatic inductor voltage. The colours".

l.262: Replace "the double" with "doubly-".

l.266: Replace "more" with "greater".

Section 2.2: This is a verbatim copy of Section 2.1 in Part 1. Please see earlier comments.

Section 2.3: Another verbatim section (which simply refers to the Part I section). These are examples that argue the two parts should be combined since it should not be necessary to read a separate paper for this information.

Eq.5: The variables in this formula are undefined. It is not sufficient to refer to a separate paper to define the variables.

Eq.6: I suggest you add a figure to show the relative importance of these two force terms – the Coulomb term and the image charge term - versus radial distance, under the experimental conditions of the present paper. If the other dynamic forces can also be indicated for comparison, so much the better.

l.326: What is the meaning of the ambiguous word "global"? If it indicates "mean" then use "mean".

l. 341: Replace "repulsing" with "repelling".

l.341: Replace "the fact that" with "whether the".

l.343-347: The short-range attractive force needs to be pointed out in Fig.7. I assume it is the small rise in CE at positive q x Q?

Fig.7: Add a vertical axis/line at q x Q = 0 so the key transition from attractive to repulsive Coulomb force can be seen.

Fig.7: State the droplet size in the caption.

Fig.7: Dark blue and black points are indistinguishable. I suggest you use different symbols for the three AP charges and then colour the points with a rainbow legend according to droplet charge.

Fig.7: Please add your theoretical curve to this figure from Dépée et al., 2019. Does it pass through your measurements? If not, please explain the discrepancies. Do you predict the small inflection in the CE as q x Q goes from negative to positive?

Fig.7: Concerning the 3 points in the bottom right hand corner, are you capable of measuring CE at 1E-4 and below? Figure 10 would suggest not. One of the points disagrees with other points at higher CE values but the same q x Q. The error bars on these 3 points look unrealistically small.

l.371-372. Please clean up this sentence.

l.374-379: You highlight the fact that these are the first experimental data to show the short-range attractive image charge forces but you do not provide any quantitative comparison in Fig.8 with your detailed model (Dépée et al., 2019). Please correct this.

Fig.8: Instead of the lines joining the points, please add curves showing the predictions from your model (Dépée et al., 2019) – including the uncertainties in the residual Coulomb force due to the 0±600e charge on the droplets.

Fig.8: State the droplet size in the caption.

Fig.9: I suggest this figure (and Fig.10) is better plotted as Measured CE (y) versus Modelled CE (x). The model (if calculated properly) has no errors but the measurements do have errors and so they are better shown on the y axis. The CE data then appear above or below (or in agreement with) the theoretical prediction. The dashed line should be labelled "Measurement = Model".

Fig.9: I don't understand what a "Parity plot" means. Better to label this "Measured versus modelled collections efficiencies according to Kraemer…".

l.435: Replace "compared" with "comparable".

l.453-470: This seems like a very long-winded paragraph that could be replaced by a brief sentence: "The lower detection limit on our experimental collection efficiencies is 1E-4" (or whatever is the correct number).

Fig.10: Please follow the same axis convention as Fig.9 (Measured CE (y) versus Modelled CE (x)).

Fig.10: Add a second curve to show the modelled collection efficiency without any charge effects so we can see the relative importance for the CE of charge compared with dynamic effects.

Fig.10: Please indicate in the caption what the error bars indicate. They are clearly underestimated and do not represent the full errors for each point. Please indicate the magnitude of the systematic errors either on a few representative points or else quoted in the caption.

Fig.10: State the experimental conditions – or their range – in the caption.

l.525: Replace "got" with "have".

l.560: Replace "considering" with "that include".

Fig.12: This is another figure that uses 3D images but would be far clearer and more useful if it were replaced by a simple line schematic. Please indicate precisely where on the new figure the image shown in Fig.13 is obtained.

---

## Author Comment (AC1) · 1 Feb 2021

Dear Professor,

Co-authors wish you a happy new year and we hope you are doing well. We first want to thank you for the delay you provided us for this answer. We also want to thank both reviewers, we found their remarks very constrictive, they enhanced the quality and clarity of the article. Please find the point by point answer to aech question.

Please, feel free to contact us for any information. Kind regards,

Pascal LEMAITRE

Please also note the supplement to this comment:
https://acp.copernicus.org/preprints/acp-2020-832/acp-2020-832-AC1-supplement.pdf
* * *
[Figure]

**Supplement:**

Review for *Laboratory study of the collection efficiency of submicron aerosol particles by cloud droplets. Part II - Influence of electric charges* by Dépéé et al. The paper presents an experimental of collection efficiency of submicron aerosol particles by cloud droplets under high humidity while testing different electric charge setups. This is a very interesting work with a very nice setup. I believe the paper fits well with ACP. My only issue is that the paper needs some organization clarification and rewriting of some parts to be suitable for publication.

**General comments**

You are citing your work part 1 before it has been published which is a bit problematic. I do not recommend using this (unless the editor disagrees) at lease until it has been accepted. You have too much depended on it in this paper which makes it hard to understand can this paper stand by itself.

I had to read your other paper (part 1) to understand this paper, as you refer to it in almost every section. You need to provide more explanation (instead of sending us to part 1) and refer to reader to the part 1 only in case they would like to read something in more detail

**It is a clear that you need to read both parts to deeply understand the work. However, we followed your comment and we added some information in this paper (the uncertainty section for example). Note that we also added the section number when we quoted the other Part in this paper so the reader can easily look for information in the other paper.**

**Yes, we are citing our other part in this one whereas no one has been accepted but it was the aim of the submission. We present our submission like this at the editor – two papers from a same work but with a lot of differences (setup, CE calculation, CE uncertainties, physical effects observed). So, we feel two papers are required instead of writing a single 50 pages paper with too many information. At the end the reader would have been completely lost about what setup or what method is applied for what experimental results. We know a parallel two submission is not that much usual but we feel it is the best way for the reader.**

I think a better organization of your experimental set up is required. As a reader, I had to jump back and forth to understand what you were referring to. Try to combine the different sections into your general set up description. You can also use fewer words to describe thins as thermodynamic conditions. You can write this information in a 1-2 sentence. Also, you can add Fig 3 and 5 as part of the setup figure with sections of Fig A, Fig B. Please, and tell us what exactly we see on the figure and where, as it is not understood. For example, I do not understand where is the location of your piezoelectric injector?

**We completely changed the Figure 1 and now you can see the piezoelectric position as well as the aerosol injection, the flowrates, the main function of the different parts. With the explanation in section 1.1 we feel it is better for the understanding.**

Does the aerosol pass the humidifier area? This means you increase their size due to the hygroscopic process, have you evaluated that aspect?

Yes, the APs pass in the humidifier (as shown in Figure 1 and explained in the first paragraph of "1.1 overview") and in the collision chamber the relative humidity is close to 95 %. So that the hygroscopicity is considered as you can see in section "2.2 collection efficiency definition' equation (2) the term $GroF$ appears. As stated in the text "$GroF$ - the growth factor depending on the relative humidity ($RH$). This latter characterises the hygroscopicity of the sodium fluorescein salt - further details related to its evaluation can be found in section 1.2.3.3 of Dépée et al. (2020)."

Note that we added a sentence in "1.1 overview" at the end of the $2^{nd}$ paragraph:

"Before the AP injection in the In-CASE's chamber, the flow is humidified to ensure a high relative humidity level inside the collision chamber (section 1.2). Thus, the hygroscopicity of the sodium fluorescein salt is considered during the experiments (see section 2.2)."

Also, what is your aerosol size distribution, you mention sizes in the result (not the method) but do not show it.

In section "1.1 Overwiew" before the result section, we stated:

"Once generated, the APs flow through a diffusion dryer and a portion of the flow is then directed into a Differential Mobility Analyser (DMA; TSI 3080) to select APs following their electrical mobilities whereas the overflow ends in an exhaust. At the DMA's outlet, the AP size distribution is assumed to be monodispersed (discussed in section 2.1)."

In the method part, you do not mention if the droplets stay stable or grow in your chamber since RH is 95%, you mention it later in the result.

Since the RH is below 100 %, the droplets evaporate but as stated in the method part "1.3 Droplet evaporation" :

"The droplet evaporation was theoretically evaluated through the section 13.2 of Pruppacher and Klett (1997). The terminal velocity of the droplet ($U_{A,\infty} \approx 25$ cm/s) is computed from Beard (1976) meanwhile the droplet residence time in the collision chamber ($\approx 4$ cm) is deduced from the changes in droplet radius and terminal velocity. For a relative humidity level of 95 %, it was found that the droplet radius decreases by less than 0.3 % from the droplet generation to the bottom of the collision chamber. Thus, the droplet evaporation in the In-CASE collision chamber was neglected for the discussions below."

Note that this section has been updated to be independent from Part I.

This led me to the last part you barely mention anything on uncertainties in the method later in the result you explain some of the differences you got by explaining them.

**The sections "1.3 Droplet evaporation" and "2.3 uncertainties" were updated to be more independent from the Part I paper. Note that we also changed the end of the section "2.2 collection efficiency definition" where the calculation of the mean AP mass concentration in the collision chamber is quite different than Part I.**

Overall, I think you should organize the paper better it feels like you are jumping between your experimental results and the model which make it's hard to follow

**We did our best to correct this impression as previously stated in the responses.**

**Specific comments Abstract**
In general, it is not recommended to use references in the abstract. Just keep it general and mention you will compare to literature or expend other works.

**We updated the end of the abstract "The measurements are then compared to theoretical models from literature – showing good agreements.".**

**Introduction**
Lines 37-38 the use of this example *developing cardiovascular disorder (Crouse et al., 2012)* is wrong, as the impact is not just cardiovascular, I recommend to keep it general - *They are also a key topic in human health where Aps increase the likelihood impacting human health (morbidity and mortality) (new citations).*

**You are completely right, we updated the sentence:**

[revised manuscript text omitted]

Lines 49 -50 can you change the sentence – this is very confusing - *leave the streamline that surrounds the falling droplet* maybe you can say that they are swept by the streamline of the droplets or something like that.

**This sentence was updated following the comment from another reviewer:**

**"To be collected an AP has to deviate from the streamline around the falling droplet to make contact with it."**

Line 51 change the word *motion*

**We changed "motion" by "trajectory"**

Line 53 change the word *massive*

**We did the choice to keep "massive".**

Line 54 use other words than *to go*

**We changed "to go" by "to move"**

Lines 61-62 instead or writing *These effects prevail in a subsaturated air - as it is the case sometimes in clouds - and are discussed in Part I (Dépée et al., 2020).* I would prefer to also see a description of them here which will make the reader life easier to understand the prices

**We deliberately omitted to explain these effects since it is exhaustively detailed in Part I. Similary we also referred the electrostatic forces in Part I without explaining them – citing Part II.**

**As you can read, in Part I we stated of thermophoresis and diffusiophoresis :**
**"Note that the electrostatic forces can have a significant influence on the CE (Tinsley and Zhou, 2015; Dépée et al., 2019). This effect will be discussed in a companion paper (Dépée et al., 2020) of this work."**

**And in part II we stated of electrostatic forces:**
**"Note that there are also thermophoretic and diffusiophoretic forces which can have an influence on the CE. These effects prevail in subsaturated air - as it is the case sometimes in clouds - and are discussed in Part I (Dépée et al., 2020)."**

Lines 68-71 I believe your example is not good, just state that AP is charged by natural atmosphere phenomenon e.g. lightning process, dust storm. The use of this example *particularly for nuclear safety issues when the APs removal by clouds result from the discharge of radioactive materials from a nuclear accident* is so low compared to all other phenomena, I do not recommend using it.

**Few authors of these two papers work for the French Institute of Radiological Protection and Nuclear Safety so we have to mention what we are paid for. We mean the main application for our Institute needs to be emphasised. Nevertheless, just before the sentence we pointed that droplets and atmospheric APs are known to be charged. So that the electrostatic forces**

**can increase the CE. Afterward, we stated that the radioactive APs are even more charged so that the "electroscavening" in clouds has to be studied. For the atmospheric APs but even more for radioactive APs which are our application.**

Besides, I think you should mention that electroscavenging is more important for smaller particle sizes (<0.1 μm) than Brownian diffusion as found by the model work of Tinsley et al. (2001) and Ardon-Dryer (2015).

**Note that Tinsley et al. (2001) do not model the Brownian motion. The authors started to study the impact of the Brownian motion on CE after 2010. Moreover, we do not recommend to use the kernel equations for the difference mechanisms summarised by Ardon-Dryer (2018). Indeed, these equations are deduced from the assumption that the microphysical effects are independent each other and it is wrong. These equations are suitable to get easily modelled CE but not appropriate to talk about the contribution of the microphysical effects involved in the CE. Indeed, it is better to consider the modelled CE from lagrangian models like Tinsley and Zhou (2015) – who model the Brownian motion – or Dépée et al. (2019). From the both works it is not that easy to state that** "electroscavenging is more important for smaller particle sizes (<0.1 μm) than Brownian diffusion"**. Indeed:**

- **When you increase the droplet radius for the same droplet and AP charges, the relative contribution of the electrostatic forces decreases for the benefit of the Brownian motion.**
- **When the droplet is neutral, the modelled CE considering the electrostatic effects tends to the one without electrostatic effects for smaller AP radii. So, when the droplet is neutral, the Brownian motion starts dominating when you decrease the AP radius. Note that, the more the AP is weakly charged, the more the Brownian motion domination appears for larger AP radius.**
- **Sometimes, when the charge product of the droplet and AP is large, the Brownian motion never dominate from the Greenfield gap to the nanometric AP radius – the electrostatic effects are of first contribution by far.**

**So, it is more complicated and we prefer to mention at the end of the 5ᵗʰ paragraph (introduction):**
**"A detailed study of their contribution can be found in Dépée et al. (2019)."**

Lines 75-90 the entire paragraph is not clear, I recommend rewriting it, also provide the eq here since it is relevant for this paper.

**We changed few things in the paragraph but not that much. Indeed, it is the simplest explanation we can give and as stated in the paragraph, more details can be found in Tinsley et al. (2000) for the analytical equation and in Dépée et al. (2019) for the contribution of the two terms in the equation: the coulomb inverse square and the attractive-short range terms.**

**Note that we did not provide the equation here since:**

- **In general, it is not recommended to put equation, figure, table in introduction;**
- **We give the equation later in the paper so we don't think it is good to write few times the same equation in the paper;**
- **It is better to have the equation in the section results when the reader truly needs it to follow the observation of the CE measurements.**

Lines 91-106 will be nice to organize this and write this in a way that will flow better, I provide below small examples to improve this paragraph. Organizing all information in a table could help the reader to understand similarities and differences between previous works.

**We don't think giving the reader the droplet/AP charge or size or other experimental conditions for these works will be useful in our demonstration. The aim here is not to provide a comparison between the experimental works but just to talk about the limit of their works mainly due to their experimental setup. Thus, we explained that their CE measurements are difficult to compare each other. Finally, we emphasised the main gap which gives importance to our paper:**

**"only the Coulomb inverse square term in the analytical expression of the electrostatic forces can be documented whereas the contribution of the short-range attractive term has not been experimentally verified until now."**

**Moreover, it is really unusual to have a table in the introduction.**

Lines 91-92 just say this instead *Several laboratory studies investigated the influence of the electric charges on CE* (citations)

**This remark is considered.**

Line 94 change *Beard (1974) do not,* to *Beard (1974) did not*

**This remark is considered.**

Ling change *Lai et al. (1978) have a polydispersed*, to *used a polydispersed*

**This remark is considered.**

Lines 111-113 please rewrite, even try to combine it with the previous sentence.

**We changed these sentences:**
**"Thus, a novel experiment has been designed to study the influence of electric charges on the CE which is presented in this paper. Note that this experiment was also used to study the influence of relative humidity which is the object of the companion paper: Part I (Dépée et al., 2020). "**

Lines 114- 124 I think is not relevant just add the comparison in the relevant parts in the paper.

**We change the last paragraph;**
**"The first part of the paper describes the experimental setup. Afterwards, the method to evaluate the CE and the uncertainties are detailed. Then, the measurements are presented and confronted with the prediction of Kraemer and Johnstone (1955) and the Lagrangian model of Dépée et al. (2019). Finally, this work concludes with the experimental validation of the Dépée et al. (2019) model and a necessary incorporation of the modelled CEs in cloud models, pollution models, climate models, and so forth, to study the "electroscavenging"."**

**Experimental Setup**
Explain some of the measurement uncertainties, as you will mention later in the result but you do not mention anything in the method part.

**We completed the section "2.3 Uncertainties" following your comment:**

**"The relative CE uncertainty ($u_{CE}$) is calculated following Lira (2003) and presented in equation (5):**

$$u_{CE} = \sqrt{u_A{}^2 + u_{H_{eff}}{}^2 + u_{N_d}{}^2 + u_{m_{AP,d}}{}^2 + u_{C_{m,AP}}{}^2} \qquad (5)$$

**Where the relative uncertainties are related to the droplet radius ($u_A{\approx}1\%$), the effective height of interaction between droplets and APs ($u_{H_{eff}}{\approx}4\%$), the number of injected droplets during the experiment ($u_{N_d}{\approx}2\%$), the measured AP mass in the droplet impaction cup ($u_{m_{AP,d}}$) and the mean AP mass concentration in the In-CASE collision chamber during the experiment ($u_{C_{m,AP}}$).**
**The relative uncertainty $u_{m_{AP,d}}$ is evaluated through the equation (6) :**

$$u_{m_{AP,d}} = \sqrt{u_{fluorimeter}{}^2 + u_{dilution}{}^2} \qquad (6)$$

**Where $u_{dilution}$ is the relative uncertainty of the dilution performed during the spectrometry analysis, assumed to be equal to 1 %, and $u_{fluorimeter}$ is the relative uncertainty of the fluorimeter which can be up to 30 % when the measured AP mass is close to the detection limit. The relative uncertainty of the mean AP mass concentration in the In-CASE collision chamber ($u_{C_{m,AP}}$) is calculated through the equations (7) :**

$$\begin{cases} u_{C_{m,AP}} = \sqrt{u_{m_{AP,tot}}{}^2 + u_{Q_{In-CASE,c}}{}^2 + u_{\Delta t}{}^2 + u_P{}^2} \approx \sqrt{u_{m_{AP,tot}}{}^2 + u_{Q_{In-CASE,c}}{}^2 + u_P{}^2} \\ u_{m_{AP,tot}} = \sqrt{u_{fluorimeter}{}^2 + u_{dilution}{}^2} \end{cases} \qquad (7)$$

**$u_{m_{AP,tot}}$ is the relative uncertainty of the measured AP mass on the HEPA filter which depends on the relative uncertainties of the dilution ($u_{dilution}{\approx}1\%$) and the fluorimeter ($u_{fluorimeter}{\leq}30\%$) - $u_{Q_{In-CASE,c}}$ is the relative uncertainty of the AP flowrate in the collision chamber equal to 1 % - $u_{\Delta t}$ is the relative uncertainty of the experiment duration which is neglected here. More details are addressed in Part I, section 2.3 (Dépée et al., 2020) where the same definitions are used, except the relative uncertainty of the AP penetration in the collision chamber ($u_P$) is added here (equation (8)).**

$$u_P = \frac{1 - P_{InCASE,a,q}}{2} \qquad (8)$$

As mentioned in 3.2.3.2, an AP pollution independent from the experiment (pollution during the spectrometry analysis, when the droplet impaction cup is extracted at the end of experiments, etc.) remains and should be considered in equation (5). Indeed, it can significantly increase the CE measurement, especially when the measured AP mass is close to the detection limit. It means, when the CE is below $10^{-4}$ for the experiment durations considered in the study. Instead of omitting this uncertainty which is difficult to evaluate, the low uncertainty for the CE measurements below $10^{-4}$ were increased until the end of the axis in Figure 7 and 10.
Also, we assume the APs have the same charge ($q$). Actually, it exists an AP charge distribution which, even slightly dispersed, can affect the CE measurement. Nevertheless, the AP charge distribution was not measured here.
"

How do you know the droplets reduced in size by 0.5% is it based on calculation or measurements?

We updated the section "1.3 Droplet evaporation", the droplet evaporation is theoretically evaluated:

"1.3    Droplet evaporation
The droplet evaporation was theoretically evaluated through the section 13.2 of Pruppacher and Klett (1997). The terminal velocity of the droplet ($U_{A,\infty} \approx 25$ cm/s) is computed from Beard (1976) meanwhile the droplet residence time in the collision chamber ($\approx 4$ cm) is deduced from the changes in droplet radius and terminal velocity. For a relative humidity level of 95 %, it was found that the droplet radius decreases by less than 0.3 % from the droplet generation to the bottom of the collision chamber. Thus, the droplet evaporation in the In-CASE collision chamber was neglected for the discussions below.
"

What do you mean by Penetration tests?

Since APs are charged, the AP deposition can be increased in the collision chamber and the pipes due to electrostatic forces. Then, penetration tests have been performed to evaluate the penetration. This is important since we correct the mean AP mass in the collection chamber due to the AP penetration.

We updated the end of the section "2.2 collection efficiency definition':

"In equation (2), the mean AP mass concentration ($C_{m,AP}$) in the In-CASE collision chamber is evaluated from the fluorescence spectrometry analysis of the HEPA filter. It is given by the equation (4) where $Q_{In-CASE,c}$ is the AP flowrate going through the In-CASE collision chamber and $m_{AP,tot}$ is the total AP mass on the HEPA filter at the end of the experiment.

$$C_{m,AP} = \left(1 + \frac{1 - P_{InCASE,a,q}}{2}\right) \frac{m_{AP,tot}}{\Delta t \times Q_{In-CASE,c}} \qquad (4)$$

The mean AP mass concentration is corrected considering the penetration ($P_{InCASE,a,q}$) in the collision chamber which depends on the AP radius ($a$) and charge ($q$). This parameter was estimated during *ex situ* experiments where the set-up was the same as Figure 2. The only difference is a Condensation Particle Counter (CPC) set after the neutraliser when the charger is switched on, and after the aerosol charger when the CPC is switched off. Thus, the measured penetration accounts for the AP deposition due to electrostatic forces on the wall of the collision chamber as well as in the pipes from the AP charger to the HEPA filter and the humidifier (Figure 2). The measured penetrations are presented in Table 2. It is observed the penetration decreases when the AP charges ($q$) increases and the AP radius ($a$) decreases since the electrical mobility of APs is larger. During experiments, the AP number concentration was ranged from $3.10^4$ cm$^{-3}$ (for $a$=100 nm and $q = -10 \pm 1\,|e|$) to $2.10^3$ cm$^{-3}$ (for $a$=250 nm and $q = -90 \pm 9\,|e|$). As a reminder, the pipes are anti-static and connected to the ground (as well as the collision chamber) so there is no accumulation charge due to AP deposition during experiments. Thus, the penetrations presented in Table 2 are assumed to be constant over time. Note that the AP deposition was neglected in Part I (Dépée et al., 2020) since the penetration was almost 100 % when the APs are neutralised.

Table 2 Measured penetration for the experimental conditions.

| Dry AP radius ($a$) | AP charge ($q$) | Penetration ($P_{InCASE,a,q}$) |
|---|---|---|
| 100 $nm$ | $-10 \pm 1\,|e|$ | $94.7\,\%$ |
| | $-20 \pm 2\,|e|$ | $86.0\,\%$ |
| 150 $nm$ | $-11 \pm 1\,|e|$ | $96.5\,\%$ |
| | $-30 \pm 3\,|e|$ | $86.2\,\%$ |
| 200 $nm$ | $-10 \pm 1\,|e|$ | $97.0\,\%$ |
| | $-34 \pm 3\,|e|$ | $88.8\,\%$ |
| | $-71 \pm 7\,|e|$ | $78.2\,\%$ |
| 250 $nm$ | $-22 \pm 2\,|e|$ | $94.1\,\%$ |
| | $-52 \pm 5\,|e|$ | $89.6\,\%$ |
| | $-90 \pm 9\,|e|$ | $81.8\,\%$ |

"

Note that we also talk about AP penetration in Appendix A. Here the penetration was investigated through the home-made AP charger. The aim was to find the geometry and the AP flowrate in the charger which maximise the penetration. Note that the AP penetration was defined at the charger outlet as the AP number concentration when the charger is switched on over the AP number concentration when this latter is switched off. We found that with our geometry (Figure 3), the penetration is optimised with an AP flowrate of 1.5 l/min. It means, we minimised the AP deposition in the AP charger for a given AP charge at the end. The penetration optimisation was important in our study since it optimised the number concentration of charged AP in the collision chamber and reduces as much as possible the experiment duration.

In Appendix A.1, we had the definition of the penetration in the AP charger :

**"Note that the AP penetration is defined, at the charger's outlet, as the AP number concentration when the charger is switched on over the AP number concentration when this latter is switched off.**

**"**

I think your information in section 2.1 and Table 1 should be in the setup section as it describes the particles you used for the experiment.

**Table 1 is the main justification of the paper Part II we validated the assumption our AP size distribution is monodispersed. We wrote this justification just before the collection efficiency calculation since the method comes from this assumption.**

**Note that the same table is presented in Part I but the assumption that the AP size distribution is monodispersed can't be formulated there. So, in Part I, the collection efficiency is deduced differently in presence of different AP sizes in the collision chamber.**

Please describe the main Uncertainties that may impact your results in this paper

**As previously answered, we completed the section "2.3 Uncertainties" following your comment.**

Lines 166 -169 you do not need this, this paper stand by itself, if you insist using the reference of part 1 at least explain how the set up here is different.

**In part I you can find pictures obtained in shadowgraphy of a droplet train and a droplet. We feel it is pretty interesting so we let the first sentence. For the last sentence, you are right we deleted it.**

**So, the end of the section is now:**
**"Further details can be found in section 1.4 of Dépée et al. (2020) but note that the size distributions of the droplets generated by the piezoelectric injector are considered monodispersed since the droplet size dispersion is very low ($\sigma \sim 1\%$)."**

Line 185-186 instead of sending us to the paper just write in short here. Here you said RH was kept at95% in the other paper as 71% so I am confused that's why you need to write it clearer here.

**We updated following your comment. The section is now independent from Part I.**

In Fig 6 you show different droplets sizes, how do you know these sizes how they were evaluated.

**It is a good question. During experiments to measure CE, the droplet radius is measured by optical shadowgraphy, note that we updated the sentence at the end of the section 1.1 :**

**"The droplet size is measured during experiments by optical shadowgraphy (with a strobe and a camera, Brown in Figure 2) through two facing windows in the injection head."**

**Now, following your question we added in the caption of the Figure 6:**
**"Note that the droplet radii were not directly evaluated by optical shadowgraphy. See Appendix B for more further details."**

**For your question, in Appendix B.2 it is stated:**
**"**

**Then, the instantaneous droplet velocity** $\overrightarrow{U_{D_0}(t)} = U_{D_0,x}\overrightarrow{u_x} + U_{\infty,A}\overrightarrow{u_y}$ **at the first detected droplet position** $(D_0)$ **of coordinates** $(x_{D_0}, y_{D_0})$ **is calculated and the vertical velocity component** $(U_{\infty,A})$ **determines the droplet radius** $(A)$ **by reversing the Beard (1976) model. Here, the circle Hough transform is not used to calculate the droplet radius like during CE experiments - see Figure 8, right from part I, Dépée et al. (2020). Indeed, the camera zoom is at the lowest to get a large field - the uncertainty would be too large.**
**"**

**Note that we changed a bit this paragraph because we never stated in this paper that the droplet radius is evaluated during CE experiments by optical shadowgraphy and with a circle Hough transform. So, we needed to quote "Figure 8, right from part I, Dépée et al. (2020)."**

Line 241 – you cite Ardon-Dryer et al. (2015) but it is not your reference list

**We updated following your comment.**

Lines 284-285 information of the duration of each experiment should have been provided in the experimental section

**Following the comment of another reviewer, we added the table 4 which gives the key feature of the experiment. There, you can find the experiment duration.**

**You can also get this information in section 2.2:**
**"Δt - the experiment duration (from 3 to 6 hours)"**

Line 285 - growth factor of what droplets or aerosol

**We changed the sentence:**
**"$GroF$ - the growth factor of AP depending on the relative humidity $(RH)$."**

**Note that since the relative humidity is below 100 % the droplet evaporates (neglected in the paper) but does not grow. This growth factor is for the AP due to the hygroscopicity of the sodium fluorescein salt. This growth is a chemical process.**

**Results and discussions**

Line 308 - Extension of the Dépée et al. (2019) model,
Maybe this should be in your method parts and nor as part of your result Lines

The two papers are experimental studies and we wanted to separate the experimental method which is the aim of the paper from the modelling. So that this section is needed in the result section but not to explain the experimental setup and the method. Then we feel it is better here. Note that it is in the first subsection, before the result subsections. It is as an aside.

312-313, unclear what do you mean?

We updated following your comment:

"Since the mean relative humidity level was 95.1 ± 0.2 % in all experiments, the thermophoretic ($F_{th}$) and the diffusiophoretic ($F_{df}$) forces were also considered for the comparison with the model. Indeed, Dépée et al. (2020) showed that the contribution of these two effects is significant even though the relative humidity is close to 100 %."

Lines 316-317 hard to follow what belongs to what, which Eq 9? perhaps add them as an appendix

We updated following the comment of another reviewer and yours:

« Thus, the Dépée et al. (2019) model is extended here by replacing the resulting velocity at the AP location ($U_{f@p}^{*}$ in their Equation 6) by the equation (9):

$$U_{f@AP}^{*}(t) = U_{f@AP}(t) + \frac{\tau_{AP}}{m_{AP}}\left(F_{buoy} + F_{df} + F_{elec} + F_{th}\right) \tag{9}$$

Where $U_{f@AP}$ is the fluid velocity at the AP location, $\tau_{AP}$ the AP relaxation time and $m_{AP}$ the AP mass. The expression of the buoyancy force ($F_{buoy}$) is detailed in equation system (B.1), $F_{df}$ and $F_{th}$ in the equations (12) of Dépée et al. (2020). $F_{elec}$ is defined in equation (10) :
"

Line 324 and Table 2- where are these 4 AP sizes come from, in the method you presented other sizes. what do you mean wet AP, you mention it here for the first time?

The wet AP radius is the radius of AP in the collision chamber which grows due to the sodium fluorescein salt hygroscopicity, depending on the relative humidity.

In Table 1 and 2, we added "Dry AP radius" instead of "AP radius" to avoid misunderstanding and in section 3.2 we added a sentence:

"The CE measurements for various charges are presented in Table 3 for the 4 wet AP radii ($a_{wet}$) considered in this study. Note that the wet AP radii are the radii of the AP, initially dry at the outlet of the DMA (Figure 2), which grow in the collision chamber to reach their equilibrium size with the relative humidity due to the sodium fluorescein salt hygroscopicity. During experiments, the AP radius increases by a growth factor (GroF) between 1.73 and 1.75 (since we actually considered the 4 mean levels of relative humidity for the 4 AP radii used in the experiments). Further details related to the calculation of the growth factor can be found in section 1.2.3.3 of Dépée et al. (2020). In Table

3, the droplet (Q) and AP (q) charges are also informed by number of elementary charges. The mean temperature was 1.08 ± 0.12°C and the mean relative humidity was 95.1 ± 0.2 % for a droplet radius of 48.5 ± 1.1 μm. Note that the wet AP density depends on the one of sodium fluorescein salt and water. The equation (1) of Dépée et al. (2020) yielded a density of 1110 kg.m-3. The key features of the experiments are summarised in Table 4."

Consider presenting Table 2 as a plot it is hard to see any connection between the variables.

The previous Table 2 (now Table 3) present the CE measurements for people who want to directly obtain them for their personal works. In a plot it would not have been that easy. Moreover, there are 4 variables: CE, droplet charge, AP charge, AP radius so it will be really complicated to regroup all of them in a single plot.

Note that the data in this table are not used for the discussion which comes next. We only discuss on the Figures 7 to 10.

Line 367 - what about the size or charge of the droplets was it kept constant? In the table, before you mention different charges, which one was used here

Following the comment of another reviewer, we add the new table 4 which gives the key feature of the setup, the experimental condition. In every caption from Figure 7 to 10 we linked this table so the reader easily goes read the experimental parameters. We also wrote the droplet radius in the 4 captions.

As mentioned in the subsection results "3.2 Collection efficiency measurements"
"The mean temperature was 1.08 ± 0.12°C and the mean relative humidity was 95.1 ± 0.2 % for a droplet radius of 48.5 ± 1.1 μm."

The droplet radius was the same during the all 70 CE measurements: 48.5 ± 1.1 μm. For the droplet charge, you are in the subsection "3.2.2 Effect of the AP charge on the collection efficiency for a neutral droplet" so the droplet is neutralised. The droplet charge is 0 ± 600 elementary charges (see the new Table 4).

Note that we updated the first sentence of the section: "In Figure 8, the CE measurements (circle) for a neutral droplet (Q=0 ± 600 |e|) […]"

Now, if you talk line 367 about "the theoretical ones" which refers about the CEs computed with the model, the parameters used were the experimental parameters (Temperature, pressure, droplet/AP radius and charges).

Line 346 - What do you mean have like signs

"The droplet and AP charges have like signs" means the droplet and AP charges have the same sign. They are both negatively charged or both positively charged.

Note that we used the same terms "have unlike or like signs" than Dépée et al. (2019) or Tinsley and Zhou (2015).

Dépée, A., Lemaitre, P., Gelain, T., Mathieu, A., Monier, M., & Flossmann, A. (2019). Theoretical study of aerosol particle electroscavenging by clouds. *Journal of Aerosol Science*, *135*, 1-20.

Tinsley, B. A., & Zhou, L. (2015). Parameterization of aerosol scavenging due to atmospheric ionization. *Journal of Geophysical Research: Atmospheres*, *120*(16), 8389-8410.

Line 349-350 – what do you mean this is not clear

We updated the sentence:
"Note that the same influence of the charge product on the CE is observed for the other three wet AP radii – the CE varies up to four orders of magnitude."

Line 362-364 – seems like something as figure caption more than information on what presented in fig 8

As every figure presentation, we remind the caption information at the beginning of the section. So, we are sure the reader will understand the Figure without necessarily go to the caption.

Line 368 -370 this is not true, you write as if forces are equal, but we know they are not. Depending on the particle size

For these AP radii, we are closed to the Greenfield gap.
Here:
- the Brownian motion is weak since we talk about AP radii of 260, 346 and 432 and the mean free path in the air is 60 nm (for 1°C and 1atm)
- the inertia of the particle is neglected since the particle radius is really smaller than 1 μm and the AP wet density is small (1110km/m$^3$)
- the thermophoresis and the diffusiophoresis contribution are comparable to the one of the Brownian motion since the relative humidity is 95 % which is close to the saturation (see Figure 10 of Part I Dépée et al. (2020)
- In figure 8, you see the three measured CEs with electric charge on AP and droplet are closed to the horizontal dashed lines which are the modelled CEs without electrostatic forces

So as stated in these sentences, we pointed that we are closed to theoretical CEs without electrostatic effects. So here, we write that for the 3 points (346 and 260 nm with -10 |e| and 432 nm with -20|e|) not only the electrostatic effect may contribute at the AP collection since all the microphysical effects appear having a small and comparable contribution each other.

**It is a qualitative observation, not a quantitative one – we did not state that are equal since we do not know that. We say the all microphysical effects have probably an equivalent contribution on the CE.**

**Finally, this sentence prevents someone from telling us "you state that you see an increase in your measurements but some data have the same value of your theoretical ones disregarding the electrostatic effects. So maybe your three data points are influenced by other effects like thermophoresis and diffusiophoresis for example. So, it is difficult to compared these points with the others significantly greater than the modelled CEs without electrostatic forces where the contribution of the electrostatic forces is of first order by far.**

Lines 398-399 – rewrite the sentence add information about why it is important to do it

**We feel like the motivations which lead us to compare the CE measurement with our model must be written in introduction and conclusion. At the middle of the paper it sounds inappropriate. "Why it is important to do it?" is stated in introduction (4th paragraph) and conclusion (1st paragraph). So, we decided to completely removed the sentence following your comment.**

Line 415 no need for *(when color goes to blue).*

**We updated following your comment.**

Lines 448-449, I disagree perhaps you have something in the model that has a bigger impact and it causes that uncertainties and disagreement with the measurements.

**The only CE measurements which significantly differ from the model have a detected AP mass in the droplet impact cup at the end of the experiment really close the detection limit of the fluorimeter. So, our assumption is a post experiment AP pollution. Our main difficulty was the underestimation of the CE uncertainties for the low CE measurements (below $10^{-4}$). Following the remark of the other reviewer, we decided to increase the low uncertainty for the CE below $10^{-4}$ since the detected AP mass is close to the detection limit. We assumed the real CE is significantly smaller but we didn't perform the same experiment with greater experiment durations since it requires more than 10 hours to get a detected AP mass significantly greater than the detection limit.**
**Nevertheless, we feel it is suitable to keep the 6 CE measurements which are probably polluted but it gives an upper limit of the true CEs.**

**What do you think in the model can cause these differences?**

**Note that in the uncertainties section, we wrote:**

"As mentioned in 3.2.3.2, an AP pollution independent from the experiment (pollution during the spectrometry analysis, when the droplet impaction cup is extracted at the end of experiments, etc.) remains and should be considered in equation (5). Indeed, it can significantly increase the CE measurement, especially when the measured AP mass is close to the detection limit. It means, when the CE is below $10^{-4}$ for the experiment durations considered in the study. Instead of omitting this uncertainty which is difficult to evaluate, the low uncertainty for the CE measurements below $10^{-4}$ were increased until the end of the axis in Figure 7 and 10.
"

And in section 3.2.3.2 "
Nevertheless, 6 data points seem inconsistent with discrepancies between model and measurements from 150 to 1000 %, occurring for the smallest CE values in Figure 10 (lower left). Note that the discrepancies should be even worse since the modelled CEs, set to $10^{-5}$, are actually much lower. By examining these data points, it appears that the measured AP masses in the droplet impaction cup - $m_{AP,d}$ in equation (1) - are very close to the detection limit of the spectrometer used. Moreover, for the experimental conditions, the model predicts AP masses in the droplets lower than the detection limit since the Coulomb inverse square term in equation (10) was very repulsive. So, the assumption can be made that a pollution occurred during the various steps of the protocol (end of experiment, disassembly of the chamber's bottom to reach the droplet impaction cup, change of room for the analysis, etc.). Note that the detection limit of the spectrometer is $10^{-15}$ kg (for the nominal analysis volume considered), which only represents ten APs with a dry radius of 250 nm deposited on the droplet impaction cup. Thus, it exists an important uncertainty in these CE measurements related to a possible contamination. This is difficult to quantify but the low uncertainties of the CE measurements below $10^{-4}$ were increased in Figure 10. To reduce this potential pollution, it would be necessary to work in a cleanroom or increase the experiment duration to avoid detection problem. However, for these data points the experiment duration was almost 6 hours (without mentioning the preparation, the purging and the cleaning durations) and, beyond this duration, stability problems of the piezoelectric droplet generator were frequent.
"

Lines 461-463 Unclear, these are also something you should talk about in the method part

Here, we stated that we got an AP pollution independent from the protocol and the method. So, it cannot be explained in the part dealing with the method since it is independent. Nevertheless, following the comment of the other reviewer, we completed the section "2.3 Uncertainties" from the method and at the end we talked about a potential AP pollution:

"As mentioned in 3.2.3.2, an AP pollution independent from the experiment (pollution during the spectrometry analysis, when the droplet impaction cup is extracted at the end of experiments, etc.) remains and should be considered in equation (5). Indeed, it can significantly increase the CE measurement, especially when the measured AP mass is close to the detection limit. It means, when the CE is below $10^{-4}$ for the experiment durations considered in the study. Instead of omitting this uncertainty which is difficult to evaluate, the low uncertainty for the CE measurements below $10^{-4}$ were increased until the end of the axis in Figure 7 and 10.
"

Lines 468 – 470 – information that should be in the method also

**We deleted in these lines "(without mentioning the preparation, the purging and the cleaning durations)" since we are right we do not talk in the method part:**
- **the preparation step where we check if the all flowrates are stable and correctly controlled. If the AP are correctly generated, as well as the droplet;**
- **the purging when we wait for the collision chamber to be completely free from APs (waiting for more than 5 durations to completely change the air volume in the chamber);**
- **the cleaning step when we clean the collision chamber, the AP charger, the droplet impaction cup, the DMA, we changed the different filters, etc.**

**These different steps exist and every experimenter knows how exhausting these steps can be, even if they are not part of the nominal operating of the experiment and are omitted in every experimental study from the literature.**

Line 472-473 I think you should start this section with this, talk about the agreement up to $10^{-3}$ and then talk about the disagreement

**Following your comment, we had a sentence at the 2nd paragraph:**
**"A good accordance between the model and the CE measurements are shown. […]"**

**The section is subdivided as follow:**
- **1rst paragraph, Figure presentation**
- **2nd, good agreement with the 66% of mean difference between model and measurements which is good for a microphysical effect which varies on several orders of magnitude;**
- **3rd, we talk about the 6 data points which seem polluted**
- **4th, we remind the agreement model/measurements is good and even better if we disregard the 6 data points stated at the 3rd paragraph. The mean difference becomes 38%.**
- **5th we finally formulate other assumptions which can explain the 38%**

**We feel it is a good organisation but if you really think it needs improvement tell us.**

Lines 478-492 you can calculate CE while changing AP sizes as suspected in the chamber their size might change, this way you could see if this can explain the differences you see

**The growth of the AP radius due to the hygroscopicity of the sodium fluorescein salt is quite instantaneous. Moreover, as stated in section 1.2.3.1 of Part I Dépée et al. (2020) : "Note that the AP flow before the injection head is also thermally set to inject APs with the same temperature as in the collision chamber." So, the AP radii can be considered as wet AP radii directly at the AP injection.**

But as you pointed here, we have a slight difference of max 4 % in relative humidity between the top and bottom collision chamber. As stated at the end of the paragraph section 3.2.3.2:
"
Another possible explanation is the differences in temperature and relative humidity between the top and the bottom chamber, respectively less than 1°C and 4 % (addressed in Dépée et al. (2020)). It could induce local discrepancies during the AP travel time in the chamber in terms of AP density and radius (through the hygroscopicity) or thermophoretic and diffusiophoretic forces which can change the likelihood of being collected by the droplets and then slightly change $m_{AP,d}$.
"
As a reminder, the mean relative humidity measured during experiments is 95.1 ± 0.2 %, the growth factor is 1.73. So, in the worst scenario, the top relative humidity was 97.1 % and the bottom 93.1 %. Thus, the growth factor can vary from 1.52 to 1.91. For a dry AP radius of 100 nm, the wet radius will be 191 nm at the top of the collision chamber, 152 nm at the bottom for a mean of 173 nm.

In fact, you suggested to computed 3 modelled CEs for the experimental condition and three AP radii: 152, 173 and 191 nm. So, the difference between the mean AP radius considered and the actual top and bottom radius is more ore less 10%.

Note that for the comparison model/measurements performed in Figures 7 to 10, three theoretical CEs are computed for every CE measurement which allow us to show uncertainty for the modelled CE as stated for example in section 3.2.3.2: "The horizontal error bars are the measurement uncertainties while the vertical ones are the extreme theoretical CE values considering the extreme droplet and AP charges (by adding or subtracting the charge uncertainties)."

Actually, we didn't just consider the charge uncertainties to compute the extreme theoretical CE values. We also consider the AP size uncertainty due to the hygroscopicity uncertainties (see Dépée et al. (2020), Part I, section 2.3.1). We omitted it in the paper since we saw that the AP size uncertainties in the computing of the extreme theoretical values were totally negligible compared to the AP and droplet charge uncertainties. Indeed, the model is hugely more sensitive to the product of charge on the CE than a variation of the AP size of few percent.

Note that the AP size uncertainties computed and considered in the extreme theoretical CE values are less than 5 % meanwhile the example stated above is 10 %.  Nevertheless, it would have not changed the domination of the droplet and AP charges uncertainties in the extreme theoretical CE values computed from what we see in the all simulation we performed (from Part I, Part II and Dépée et al. (2020)).

Finally, we did the choice to just list a potential change in AP radius in the collision chamber which can affects the measured CE instead of performing a sensitivity study. It comes from the fact than doing simulations with the Dépée et al. (2020) model require weeks of computational time to get theoretical values statistically reliable.

**Conclusion**
Your conclusion looks more like a discussion then a conclusion, I recommend organize it differently.

The conclusion is subdivided as followed:

- **1st paragraph : we remind the aim of the study. Why is it important to study the influence of electric charge on CE**
- **2nd, experimental conditions**
- **3rd, summary of the observation from the measurements**
- **4th, summary of the comparison measurements/prediction of Kramer and Johnstone (1955)**
- **5th, summary of the comparison measurements/dépée et al. (2019) model**
- **6th, conclusion of the present experimental work and perspectives of a numerical study of the in-cloud electroscavenging with a cloud model (DESCAM, Flossmann, 1985)**

Note that the perspectives in the last paragraph introduce a future 4th papers where the modelled CE from Dépée et al. (2019) are incorporated to the DESCAM model (Detailed Scavening Model) since the model is validated with the experimental studies (Part I and Part II). Here, we will study the impact of the new modelled CE considering the electrostatic forces on the AP scavenging.

Nevertheless, after reading the conclusion you are right than a part of the 2nd paragraph should be set in the discussion. We updated this paragraph:

"In the new measured CE dataset, the APs and droplets are accurately charged through custom-made droplet and AP chargers detailed above. Since both charge polarities are found in clouds (Takahashi, 1973), the droplets were negatively as well as positively charged during experiments. Moreover, several amounts of elementary charges on the droplet were considered to represent a neutral droplet but also the weakly and strongly charged droplets respectively found in stratiform and convective clouds (Takahashi, 1973). The AP charge varied from zero to -90 ± 9 elementary charges depending on the AP size to represent different amounts of elementary charges encountered in the atmosphere, particularly the ones of radioactive APs. The relative humidity was maximised in this experimental work (95.1 ± 0.2 %) with a mean temperature in the collision chamber of 1.08 ± 0.12°C, stable and comparable with the study of the companion paper: Part I (Dépée et al., 2020). Thus, the thermophoretic and diffusiophoretic contributions on the CE measurements were reduced as much as possible. Nevertheless, since Dépée et al. (2020) measured a significant contribution for a comparable relative humidity level, these two forces were still added to the Dépée et al. (2019) model for a reliable model/measurement comparison. Finally, the droplet radius was 48.5 ± 1.1 μm and 4 wet AP radii were used - from 175 ± 3 to 432 ± 5 nm. Note that the hygroscopicity of the sodium fluorescein salt was considered in the calculation of the wet AP radius and the AP density."

The following part was added in section "3.1 Extension of the Dépée et al. (2019) model, right after the equation (10):
"

Note that radioactive APs are known to get positively charged (Clement and Harrison, 1992) whereas the APs were negatively charged in this work (Figure 4), through the charging regime used in the AP charger (for integrity of the tungsten wire over time). Nevertheless, since we have the relation $F_{elec}(q, -Q) = F_{elec}(-q, Q)$ in equation (10), the CE measurements with the same $\frac{q}{Q}$ ratios are equivalent, assuming this analytical expression is validated by the measurements (see section 3.2.3).
"

**Reference**
For forgot to put your 2019 paper

**We added the reference.**

For your 2020 paper, you can at least write submitted and to where, since it is ACP it should have a DOI number

**We are sorry if you took time to find the other submitted papers. It is a bit too late now. If the papers are accepted, the complete reference will be informed.**

**Comments on Figures**
Figure 1.
I am not a big fan of Fig 1. It is too complex and confusing, too much information (AP and droplets size, flow charge, etc). Since you are describing, in general, the different mechanisms that impact the CE processes, I would prefer to see something more general as Fig 1. in Ardon-Dryer (2015).

**Following your comment and the comment of the other reviewer. We completely changed the Figure 1. We deleted the 3D view, added the flowrate, added a colour code for the main component of the setup, draw the piezoelectric injector and the AP injection. It sounds by far clearer, isn't?**

Figure 2.

[Figure]

What is this part for: and where is the piezoelectric injector it is not clear from the figure?

**We updated Figure 2 so now we think it is clearer. You can see the piezoelectric injector (referred as droplet generator). This part is, in the new Figure 1, the grey color. Since the AP flowrate is limited in the DMA and the AP charger is optimised for an AP flowrate of 1.5 l/min, we need to add clean air to get the required flowrate in the AP charger. It dilutes the AP number concentration but it was considered.**

**Note that in the section "1.1 Overview" you can read:**

**"Since the optimised AP flowrate in the charger is 1.5 l/min and the maximum AP flowrate in the DMA was 1.2 l/min during the experiments, a clean air flowrate is added at the charger's inlet"**

**And in the new caption of Figure 2, it appears:**

**"Grey – clean air adding for a constant flowrate at the AP charger inlet. Brown"**

Figure 5 is very similar to a figure 5 you have in a paper in part 1 do you need it here.

**You are right but some panels change between the two figures. We wanted these figures here because:**

**-part I we talked about the electrostatic inductor for the droplet neutralisation and the housing made with a 3D printer. The figure 5 there is required.**

**-Part II the mainly issue here was to show how the electrostatic inductor is placed in the injection head. The injection head is only shown here in Part II. It avoids the reader to jump between papers even if it is necessary sometimes.**

Figure 9 what size of AP was used here?

**The 4 AP sizes were used here. We selected from the 70 measurements the ones with negative charge product.**

**We updated the sentence after the equation (11):**

**"Since this prediction models the contribution of the attractive Coulomb forces on the CE, only the CE measurements with a negative charge product for the 4 AP radii are compared."**

**Following your comment, we updated the caption:**

**"Modelled CE from the prediction of Kraemer and Johnstone (1955) as a function of the measured CE. The droplet radius is 48.5 ± 1.1 μm. Only the negative charge products for the 4 AP radii are considered here, represented by the color code. The experimental conditions are summarised in Table 4."**

Figure 10 - *The color code referrers to the droplet radius*, I believe you meant AP right not droplets

**You are completely right, thanks! We updated.**

Appendix
I believe the numbers of the appendix should not continue you should give them new numbers as the appendix is a different part of the paper. Also, it would have been nice to see parts of the appendix as part of the paper when you explain your set up so we can understand in depth what you have done.

**We updated following your comment. We restarted the equation and figure numbering.**

**Initially, the parts in Appendix were in the main paper. But after the first reading before the submission, it appeared that the setup part in the paper was too long, complicated, hard to read. The general comment before the Appendix existed was "we need to read too many pages of setup and method before finally reach the CE measurements".**

**Since the *ex situ* experiments performed to get the charging low of the AP charger (Appendix A) and the charging low of the droplet electrostatic inductor (Appendix B) were not used in the nominal operating of the experiment, we decided to describe them in Appendix. Indeed, it is not necessary to**

**read these parts to understand the general setup. Nevertheless, if the reader wants to understand the AP and droplet charging and in depth what we have done, the Appendix are still here.**

Reference used in this review

Ardon-Dryer, K., Huang, Y.-W., and Cziczo, D. J.: Laboratory studies of collection efficiency of sub-micrometer aerosol particles by cloud droplets on a single-droplet basis, Atmos. Chem. Phys., 15, 9159–9171, https://doi.org/10.5194/acp-15-9159-2015, 2015.

Tinsley, B. A., Rohrbaugh, R., and Hei, M.: Electroscavenging in clouds with broad droplet size distributions and weak electrification, Atmos. Res., 59, 115–135, 2001.

---

## Author Comment (AC2) · 1 Feb 2021

Dear Professor,

Co-authors wish you a happy new year and we hope you are doing well. We first want to thank you for the delay you provided us for this answer. We also want to thank both reviewers, we found their remarks very constrictive, they enhanced the quality and clarity of the article. Please find the point by point answer to your review review.

Please, feel free to contact us for any information. Kind regards,

Pascal LEMAITRE

[Figure]

Please also note the supplement to this comment:
https://acp.copernicus.org/preprints/acp-2020-832/acp-2020-832-AC2-supplement.pdf

**Supplement:**

**Reviewer #2 comments**
Dépée et al., *"Laboratory study of the collection efficiency of submicron aerosol particles by cloud droplets. Part II - Influence of electric charges"*.

- For information, this reviewer is also a reviewer #2 of the companion paper by Dépée et al.: *"Laboratory study of the collection efficiency of submicron aerosol particles by cloud droplets. Part I - Influence of relative humidity"*.

- The present paper presents a measurement of the collection efficiency of aerosol particles by sedimenting raindrops under various electrical charge states of the raindrops and aerosol particles. The electric force between a charged aerosol particle and a charged raindrop has two components: 1) the long-range Coulomb force between charges and 2) the short-range force due to an induced image charge distribution on the raindrop (independent of its net charge state) by the charged aerosol. The latter force is always attractive and dominates when the aerosol is within a few droplet radii of the droplet surface, whereas the former dominates at greater distances and is either attractive (for unlike charges) or repulsive (for like charges). These mechanisms are collectively known as electroscavenging of aerosol particles.

- Cloud droplets – even in warm clouds – are frequently charged. Moreover, aerosol particles also may become charged at the few 10e levels due to the evaporation of charged droplets or space charge effects at the edge of cloud layers in Earth's electric field. So electroscavenging is likely to be an important microphysical process for atmospheric aerosol and cloud microphysics. However, although it has been fairly extensively treated theoretically, there are very few experimental data on electroscavenging. There was considerable experimental effort on the subject in the 1970-80's, but with limited experimental control and resulting measurements that differed by several orders of magnitude.

- The experimental measurements reported in this paper are therefore essentially unique. The experiments are carefully executed. The authors have demonstrated in Part I of their laboratory study that their experimental apparatus is capable of precise measurements of the raindrop-aerosol capture efficiency (CE) (in the former case, measuring the effects of thermophoresis and diffusiophoresis). The authors have already published a theoretical model that includes electroscavenging (Dépée et al., 2019), which agrees well with a similar model previously reported by Tinsley and co-authors. As for the Part I paper, I have no hesitation to recommend that the Part II paper also be published in ACP, after responding to the comments below.

**Thank you for this general comment**

**General comments**

- I recommend that the authors apply the same general comments that I made for the Part I paper also to the present paper. In particular the authors should ask a native English speaker to edit the manuscript for poor English grammar and lengthy sentences.

**We did our best to enhance the English writing**

- There is a lot of cross-referencing with the Part I paper, and many parts (main apparatus description, derivation of the collection efficiency, …) where texts (and some figures) are repeated

almost verbatim. I leave it to the editor and authors to decide if it is best to keep these as two separate papers or to make the Part I and Part II papers as two large chapters of the same paper.

We totally agree. Indeed, both articles are strongly linked and the reader needs to read both articles to deeply understand all the experimental details. Nevertheless, we did our best to remind the main information required to understand each article. The few verbatim are useful to avoid the readers to pass from an article to the other. Note that we added the subsection with the quote so that the reader can easily look for the information in the other paper.

Besides, we previously wrote a single paper but it was too long and completely unclear. Indeed, we really felt it is necessary to separate paper. The main reason is that the method of CE measurement (and uncertainties) are completely different between Part I and Part II. Moreover, as it is frequently the case in the literature, we preferred to focus on the influence of only one parameter in each article:

1. Pranesha, Kamra, (1996, 1997a, 1997b).
2. Wang, Pruppacher, (1977) & Wang, *et al.* (1978).

Finally writing a single article would take around 50 pages.

Pranesha, T. S., & Kamra, A. K. (1996). Scavenging of aerosol particles by large water drops: 1. Neutral case. Journal of Geophysical Research: Atmospheres, 101(D18), 23373-23380.

Pranesha, T. S., & Kamra, A. K. (1997a). Scavenging of aerosol particles by large water drops: 3. Washout coefficients, half-lives, and rainfall depths. Journal of Geophysical Research: Atmospheres, 102(D20), 23947-23953.

Pranesha, T. S., & Kamra, A. K. (1997b). Scavenging of aerosol particles by large water drops: 2. The effect of electrical forces. Journal of Geophysical Research: Atmospheres, 102(D20), 23937-23946.

Wang, P. K., & Pruppacher, H. R. (1977). An experimental determination of the efficiency with which aerosol particles are collected by water drops in subsaturated air. Journal of the Atmospheric Sciences, 34(10), 1664-1669.

Wang, P. K., Grover, S. N., & Pruppacher, H. R. (1978). On the effect of electric charges on the scavenging of aerosol particles by clouds and small raindrops. Journal of the Atmospheric Sciences, 35(9), 1735-1743.

- The most surprising aspect of the paper is that the authors have made some unique and impressive measurements *and* have developed a careful theoretical model yet there is only one figure (Fig. 10) where they compare their measurements with their model. Why do they omit any comparison of their model with their data in the other two figures (Figs. 7 and 10) where they are presented? Without this comparison it is hard to get confidence that their data do indeed verify the short range image force attraction. (I will pick this point up below.)
-
You are right, we updated the figures by adding the modelled CEs. See response for the specific comments.

**Specific comments**

l.27: Replace "the neutralisation" with "zero". We updated following your comment.

l.30: Replace "correlation of Kraemer and Johnstone (1955)" with "prediction of Kraemer and Johnstone (1955)" (and elsewhere in the text). **We updated following your comment.**

l.34: Replace "on" by "on".
**The end of the abstract was updated following the comment of the other reviewer.**

l.49 Replace "is mainly depending" with "mainly depends".
**We updated following your comment.**

l.71-74: Please clean up this sentence.
**The sentence becomes:**
**"For this purpose, the modelled CEs with electrostatic forces need to be experimentally validated before the incorporation in cloud models. Especially, the analytical expression for electrostatic forces used in numerical studies (Jaworek et al., 2002; Tinsley et al., 2006 ; Tinsley and Zhou, 2015 ; Dépée et al., 2019) has to be confirmed by measurements."**

l.114-115: Please clean up this sentence.

**The sentence becomes:**
**"The first part of the paper describes the experimental setup. Afterwards, the method to evaluate the CE is detailed."**

l.121: Remove "a".
**We updated following your comment.**

Fig.1: See the comments I made for Part I. I suggest you add some more trajectories to panel D that show aerosol particles outside the geometrical path of the raindrop, which are attracted into a collision (or not).

**(We did the same correction in Part I and Part II, see respond in Part I.)**
**We added trajectories outside the cross section of the droplet. Note that all trajectories are collected and we didn't show uncollected trajectories. Indeed, the CE is 27 for these parameters (Dépée et al. 2019) and if you neglect the Brownian motion here you have CE=xmax\*xmax where xmax is the max distance from the vertical droplet axis when the AP is collected due to the electrostatic forces (Dépée et al. 2019). So xmax=5.2 and we didn't want to plot until this value on the X or Y axis to see uncollected AP trajectories. We wanted the same scaling between the panel E and D.**

l.141: Indicate the AP (air) flow velocity and transfer time in the chamber.

**OK, note that we added another sentence.**

**"Droplets fall at their terminal velocity (≈25 cm/s) into a chamber through an AP flow of 1.5 l/min. The flow velocity is 1.3 cm/s and the AP transfer time in the collision chamber is almost 80 s."**

l.144: Replace "go out" with "pass out of".
**We updated following your comment.**

Fig. 2: You do not show in Part II but we see from Part I that the APs are introduced into the sheath region of the laminar flow down the tube, whereas the droplets fall down the centre of the tube. The APs have charges between 25e and 150e. What is their number concentration (cm-3)? I could not find this anywhere in the text and it is an important number. The APs will form a space charge in the tube that pushes them to the walls and away from the central region where the droplets fall. How big is this effect? Does it influence your estimate of the AP number concentration seen by the falling droplet?

**We updated the Figure 2 according to your comment in the review of Part I. We also updated the Figure A.1.**

**Now for the AP penetration, you are completely right we didn't explain about that but we performed tests to evaluate the penetration. It was finally corrected in the AP number concentration. Note that the penetration is evaluated from the outlet to the AP charger to the outlet to the neutraliser (Figure 2 part II). We assumed the AP deposition was the same through the chamber even if it can be larger at the AP injection or in the AP/droplet separator.**

**In section 2.2 where we define the equation for the CE, we changed the end:**
**"**

$$C_{m,AP} = \left(1 + \frac{1 - P_{InCASE,a,q}}{2}\right) \frac{m_{AP,tot}}{\Delta t \times Q_{In-CASE,c}} \qquad (4)$$

**The mean AP mass concentration is corrected considering the penetration ($P_{InCASE,a,q}$) in the collision chamber which depends on the AP radius ($a$) and charge ($q$). This parameter was estimated during *ex situ* experiments where the setup was the same as Figure 2, the only difference being a Condensation Particle Counter (CPC) set after the AP neutraliser and the AP charger to measure two AP number concentrations - 1 and 2, respectively. The penetration is then defined as concentration 1 over concentration 2. Thus, the measured penetration accounts for the AP deposition due to electrostatic forces on the wall of the collision chamber as well as in the pipes from the AP charger to the HEPA filter and the humidifier (Figure 2). The measured penetrations are presented in Table 3. It is observed the penetration decreases when the AP charges ($q$) increases and the AP radius ($a$) decreases since the electrical mobility of APs is larger. During experiments, the AP number concentration was ranged from $3.10^4$ cm$^{-3}$ (for $a$=100 nm and $q = -10 \pm 1\,|e|$) to $2.10^3$ cm$^{-3}$ (for $a$=250 nm and $q = -90 \pm 9\,|e|$). As a reminder, the pipes are anti-static and connected to the ground (as well as the collision chamber) so there is no charge accumulation due to AP deposition during experiments. Thus, the penetrations presented in Table 3 are assumed to be constant over time. Note that the AP deposition was neglected in Part I (Dépée et al., 2020) since the penetration was almost 100 % when the APs are neutralised.**

Table 3 Measured penetration for the experimental conditions.

| Dry AP radius ($a$) | AP charge ($q$) | Penetration ($P_{InCASE,a,q}$) |
|---|---|---|
| 100 $nm$ | $-10 \pm 1\,|e|$ | 94.7 % |
| | $-20 \pm 2\,|e|$ | 86.0 % |
| 150 $nm$ | $-11 \pm 1\,|e|$ | 96.5 % |
| | $-30 \pm 3\,|e|$ | 86.2 % |
| 200 $nm$ | $-10 \pm 1\,|e|$ | 97.0 % |
| | $-34 \pm 3\,|e|$ | 88.8 % |

| | | |
|---|---|---|
| | $-71 \pm 7 \, |e|$ | 78.2 % |
| | $-22 \pm 2 \, |e|$ | 94.1 % |
| 250 nm | $-52 \pm 5 \, |e|$ | 89.6 % |
| | $-90 \pm 9 \, |e|$ | 81.8 % |

"

**As you can see, here the CE calculation is not the same as Part I.**

Fig.2: You estimate the mean AP number concentration in the main tube with the HEPA filter. Charged aerosol will have higher losses in the main tube and in the pipes leading to the neutralizer. How big is this loss and is it corrected for when estimating the mean AP number concentration in the main tube?

**The questions are addressed in previous answer. The penetration was evaluated from the outlet of the AP charger to the outlet of the AP neutraliser (Figure 2 part II). We assumed the AP deposition was the same during the transfer in the In-CASE chamber, pipes, etc. As a reminder, the pipes are anti-static and connected to the ground so there is no accumulation charge during experiments.**

**Note that, during the penetration tests, we didn't notice significant AP deposition in the anti-static pipes. So that, the APs mainly deposit in the collision chamber. It was also observed when we cleaned the chamber, there was more fluorescein on the wall when we measured CE with electric charges on APs than when the APs were neutralised.**

l.175: Replace "the atmospheric one" with "one atmosphere".
**We updated following your comment.**

l.181: Replace "So," with "In this way".
**We updated following your comment.**

l.182: Replace "was" with "were".
 **We updated following your comment.**

l.198: replace "get" with "are".
**We updated following your comment.**

l.210: Replace "variating" with "varying".
**We updated following your comment.**

Fig. 5: This is a very poor figure. The "3D printing" is a black blob with no detail. It conveys no information. And what is a "3D printing"? If you mean that the piezoelectric droplet generator is installed in a housing made with a 3D printer, then state that.  I suggest a simple line schematic cross section should be provided to replace the three objects in this figure.

**In this Figure, we updated the mid panel. We replaced it by a cross section of the housing made with a 3D printer. Note that we also updated the Figure 9 right of Part I.**

**We saved the other panels since it gives few points of view to understand the setup. Moreover, the injection head is presented here, the only time in this paper. We replace '3D printing' in the two papers by "housing made with a 3D printer".**

Fig. 6: Replace "Charging low of the electrostatic inductor colors" with "Droplet charge versus electrostatic inductor voltage. The colours".
**We updated following your comment.**

l.262: Replace "the double" with "doubly-".
**We updated following your comment.**

l.266: Replace "more" with "greater".
**We updated following your comment.**

Section 2.2: This is a verbatim copy of Section 2.1 in Part 1. Please see earlier comments.
**This is the definition of the collection efficiency (the purpose of the article) so we feel it is essential to remind the definition**.

Section 2.3: Another verbatim section (which simply refers to the Part I section). These are examples that argue the two parts should be combined since it should not be necessary to read a separate paper for this information.
**We initially wanted to write only one article and, in this section, we focus on the differences between the articles:**
**Following the comment of the other reviewer, we reminded the calculation of the uncertainties in this section – we do not just refer to the Part I section. In this new section, we emphasise the difference with Part I when another relative uncertainty is added in the calculation – the relative uncertainty of the AP penetration in the In-CASE collision chamber. Since it is neglected in Part I, it is another difference between the two papers. We also mentioned the potential AP pollution during the experimental protocol which can change our results and explain some difference model/measurements. Because this uncertainty is difficult to precisely quantify, we increase the low uncertainty for the CE measurements smaller than $10^{-4}$.**

Eq.5: The variables in this formula are undefined. It is not sufficient to refer to a separate paper to define the variables.
**We updated following your comment. We defined the variables in the two papers.**

Eq.6: I suggest you add a figure to show the relative importance of these two force terms – the Coulomb term and the image charge term - versus radial distance, under the experimental conditions of the present paper. If the other dynamic forces can also be indicated for comparison, so much the better.
**In Figure 2 from Tinsley et al. (2000, see below) you can find the following figure (P parameter vs the normalised radial distance) where the P parameter is the addition of the short-range attractive term and the coulomb inverse square term in Equation (10) of Part II. As you can see, at low distance AP/droplet the sign is minus since the short-range attractive term dominate. At**

large distance the coulomb inverse square term dominates since the sign is negative or positive following the charges have unlike (K=Q/q <0) or like signs (K=Q/q >0), respectively.

Nevertheless, it is not because it is always attractive at low distance following the electrostatic forces that the AP will be collected. Indeed, it is a balance between a lot of effects – Brownian motion, inertia, airflow around the droplet, drag forces, electrostatic effects, thermo and diffusiophoresis. Note that the dynamical effects as well as the Brownian motion do not only depend of the radial distance but also the orthoradial component. Thus, it would have been too complicated in one single figure to completely understand the different contributions for the experimental conditions. Moreover, we cannot plot the same curves as Tinsley et al. (2000) considering our experimental conditions since we have almost 70 values of the K parameter (through 10 AP charges and 7 droplet charges).

That's why, we did the choice to give the simplest explanation of the two terms in the electrostatic forces in introduction. As stated in this paragraph, a complete theoretical study of the different contributions in the AP collection can be found in Dépée et al. (2019).

Paragraph from the introduction:

"Finally, the analytical expression of the electrostatic forces is the addition of two Coulomb forces between the AP and the two-point charges inside the droplet. The factored expression can be found in equation (10) and further details can be found in Tinsley et al. (2000). It consists of two terms. The first one is the Coulomb inverse square term which prevails in the AP collection for large enough AP electrical mobilities or electric charge products ($q \times Q$), attractive (Figure 1, D) or repulsive (Figure 1, E) depending on whether the AP charge ($q$) and the droplet charge ($Q$) have unlike or like signs. The second term is referred to the short-range attractive term and dominates for weak electric charge products or for small AP electrical mobilities (Figure 1, F) and is always attractive (due to the charge distribution at the droplet surface with opposite sign to the AP charge ($q$)). A detailed study of their contribution can be found in Dépée et al. (2019). "

Tinsley, B. A., Rohrbaugh, R. P., Hei, M., & Beard, K. V. (2000). Effects of image charges on the scavenging of aerosol particles by cloud droplets and on droplet charging and possible ice nucleation processes. *Journal of the atmospheric sciences*, *57*(13), 2118-2134.

[Figure]

**Figure 2 from Tinsley et al. (2000)**

l.326: What is the meaning of the ambiguous word "global"? If it indicates "mean" then use "mean".
**We indicated "mean".**

l. 341: Replace "repulsing" with "repelling".
**We updated following your comment.**

l.341: Replace "the fact that" with "whether the".
**We updated following your comment.**

l.343-347: The short-range attractive force needs to be pointed out in Fig.7. I assume it is the small rise in CE at positive q x Q?
**We added the 4 curves for the 4 AP charges and the uncertainty range for the droplet and AP charges.**
**We updated the first paragraph of this section:**
**"The CE measurements for a wet AP radius of 432 nm are presented in Figure 7 as a function of the product of the droplet (Q) and AP (q) charges. The measurements are compared to the Dépée et al. (2019) extended model (solid line) for the 4 AP charges, considering the AP and droplet**

charge uncertainties. There is a good agreement between model and measurements which indicates that the analytical expression of the electrostatic forces (equation (10)) reliably describes the observations.
Indeed, an important charge influence is measured […]"

Yes you are completely right, the small rise in CE at positive charge product is due to the short-range attractive force as stated at the end of the section :

"For small positive charge products (approximately $0 \leq q \times Q \leq 10^6 \, |e| \times |e|$), an increase of CE with a factor of more than three is measured compared to the theoretical CE value without electrostatic forces. This fact truly emphasises the contribution of the short-range attractive term in equation (10) which attracts the APs toward the droplet even though the droplet and AP charges have like signs. Indeed, as previously stated, this term prevails for small charge products (Dépée et al., 2019)."

Fig.7: Add a vertical axis/line at q x Q = 0 so the key transition from attractive to repulsive Coulomb force can be seen.

We updated following your comment.

Fig.7: State the droplet size in the caption.

We updated following your comment.

As the paper Part I, we added a table 5 for the key features of the In-CASE set-up. We also mentioned the table in the caption to get the experimental conditions.
"measurement as a function of the product of the droplet ($Q$) and AP ($q$) charges for the wet AP radius of 432 nm and a droplet radius of 48.5 ± 1.1 μm. The experimental conditions are summarised in Table 5. Color code informs about the AP charge. The dashed line represents the theoretical CE value disregarding the electrostatic forces (given the air parameters $1°C$, 1 atm, 95% of relative humidity). The solid line is the interpolation of the model (with the charge uncertainty range) for the respective CE measurements at a given AP charge."

Fig.7: Dark blue and black points are indistinguishable. I suggest you use different symbols for the three AP charges and then colour the points with a rainbow legend according to droplet charge.

We changed the blue color by the green color.

Fig.7: Please add your theoretical curve to this figure from Dépée et al., 2019. Does it pass through your measurements? If not, please explain the discrepancies. Do you predict the small inflection in the CE as q x Q goes from negative to positive?

We added the 4 curves for the 4 AP radii and the uncertainty range for the droplet and AP charges. As you can see, there is a good agreement between measurements and model and the small inflection that you describe appears in the modelling.

Fig.7: Concerning the 3 points in the bottom right hand corner, are you capable of measuring CE at 1E-4 and below? Figure 10 would suggest not. One of the points disagrees with other points at higher CE values but the same q x Q. The error bars on these 3 points look unrealistically small.

The In-CASE setup is capable of measuring CE at 1E-4 and below. The only thing you have to do is to extend the duration of experiments. For these three points, the duration of experiment was almost 6 hours (with a concentration of about $10^4$ part.cm$^{-3}$).

In Figure 10 the 6 data point which are not suitable are probably due to AP pollution in the droplet impact cup during the fluorescein analysis (take off the cup, change the room with the cup to analyse, etc). The truth is, the AP pollution remains the larger error at low CE measurements but it is really difficult to account for this pollution in the error bar calculation. We were not able to quantify this random pollution in our setup. It can be due to many things, air quality in the room where we analysed the fluorescein, pollution of beakers, syringes, etc..

Even if it is often the case, this is not because the measured CE is lower that the corresponding errors is greater. Without AP pollution, the error is mainly due to the fluorescein analysis and the two diodes corresponding to two scaling and precision. Indeed, in the fluorescein analyser you have two diodes, one giving more precision at low detected concentration (= mass for a given volume control) which quickly saturate when the concentration increases; the second with low precision at low concentration but which saturates for larger concentration than the first one. When you are near the limit of a diode precision, here you have a large error (until 30 %). But sometimes you have a larger error with a great measured mass since you are not using the same diode. Thus, you cannot only consider the CE value to state that the error has to be great or low. It is all about detected mass, experiment duration and diode used.

As a reminder, we stated in 3.2.3.2 :

"Note that the detection limit of the spectrometer is $10^{-15}$ kg (for the nominal analysis volume considered), which only represents ten APs with a dry radius of 250 nm deposited on the droplet impaction cup. Thus, it exists an important uncertainty in these CE measurements related to a possible contamination. This is difficult to quantify but the low uncertainties of the CE measurements smaller than $10^{-4}$ were increased in Figure 10. To reduce this potential pollution, it would be necessary to work in a cleanroom or increase the experiment duration to avoid detection problem. However, for these data points the experiment duration was almost 6 hours (without mentioning the preparation, the purging and the cleaning durations) and, beyond this duration, stability problems of the piezoelectric droplet generator were frequent."

l.371-372. Please clean up this sentence.

We update the sentence :

"However, at a given AP radius, an increase of the CE is observed when the number of elementary charges on the APs is larger. As a reminder, this increase appears even though the droplet is neutral (or very weekly charged considering the charge uncertainty of 600 elementary charges)."

l.374-379: You highlight the fact that these are the first experimental data to show the short-range attractive image charge forces but you do not provide any quantitative comparison in Fig.8 with your detailed model (Dépée et al., 2019). Please correct this.

See next response.

Fig.8: Instead of the lines joining the points, please add curves showing the predictions from your model (Dépée et al., 2019) – including the uncertainties in the residual Coulomb force due to the 0±600e charge on the droplets.

We deleted the lines joining the points according to your comment. Afterward, we added the modelled CE for every CE measurement. We did not interpolate the modelled CE because it would have been too busy and we did not perform enough simulations to get reliable interpolations (and performing other simulations need few weeks to be statistically reliable as stated in Dépée et al. (2019)).

The vertical error bar for the modelled CE consider the AP and droplet charge uncertainties as mentioned in the new caption.

At the end of the 2$^{nd}$ paragraph (3.2.2) we added a sentence :
"Here, the good agreement between measured (circle) and modelled (triangle) CEs confirms that the analytical expression of the short-range attractive term in equation (10) is reliable".

Fig.8: State the droplet size in the caption.

As the paper Part I, we added a table 5 for the key features of the In-CASE set-up. We mentioned the table in the caption to get the experimental conditions. The new caption is :

"
CE measurement (circle) as a function of the AP charge (q) for the 4 wet AP radii (Color code). The respective modelled CEs are also presented (triangle). The droplet is neutral with a radius of 48.5 ± 1.1 μm. The experimental conditions are summarised in Table 5. The dashed line represents the theoretical CE value disregarding the electrostatic forces (given the air parameters 1°C, 1 atm, 95% of relative humidity). The vertical error bars for the modelled CEs consider the AP and droplet charges uncertainties. "

Fig.9: I suggest this figure (and Fig.10) is better plotted as Measured CE (y) versus Modelled CE (x). The model (if calculated properly) has no errors but the measurements do have errors and so they are better shown on the y axis. The CE data then appear above or below (or in agreement with) the theoretical prediction. The dashed line should be labelled "Measurement = Model".

Since the scaling is smaller on the Y-axis than the X-axis, it is better to see the measurement and the corresponding errors on the X-axis. If we switch on the Y-axis the error bar will be smaller.

In the figure there are errors for the modelling because, as explained in 3.2.3.1 and 3.2.3.2 :
"The horizontal error bars are the measurement uncertainties while the vertical ones are the extreme theoretical CE values considering the extreme droplet and AP charges (by adding or subtracting the charge uncertainties).".

Indeed, the uncertainties for the droplet and AP charges does not appear in the CE calculation so that they do not appear in the errors of the measurements. Nevertheless, these uncertainties exist. At the end of the calculation, the measured CE can be the one of the droplet and AP charges predicted as well as the charges more or less the uncertainties. Thus, we modelled the CE for the experimental conditions as well as the min and max charges to get the max and min CE modelling which can correspond to the measured CE.

On the figure 9 and 10, note that we deleted the label of the droplet radius and we added a table for the experimental condition of these experiments (as you recommended on the other paper).

Fig.9: I don't understand what a "Parity plot" means. Better to label this "Measured versus modelled collections efficiencies according to Kraemer…".

**We deleted "Parity plot" in the text and the captions.**

l.435: Replace "compared" with "comparable".
**We updated following your comment.**

l.453-470: This seems like a very long-winded paragraph that could be replaced by a brief sentence: "The lower detection limit on our experimental collection efficiencies is 1E-4" (or whatever is the correct number).
**There is no clear limit of CE calculation in the present experiment. As far as the separation of droplet and aerosol is extremely efficient (100 % for the experiment duration considered in the study) the limit is not on the CE but on the mass collected in the impaction cup. It is possible to increase this mass by increasing the duration of the experiments. However as observed from figure 7, collection efficiencies seem to highlight vertical asymptotes, that would induced very long experiments for a low increased precision. For 6 data points, we suspect a post experiment contamination of the cup that induces overestimation of the collection efficiency). In present study we limited to 6 ours experiments. To represent this, we extended the lower limit of the uncertainties for the collection efficiencies bellow $10^{-4}$. It would have been possible to increase accuracy with longer experiment (40 hours) or if the entire experimental setup were in a clean room in order to avoid this post experiment contamination.**

Fig.10: Please follow the same axis convention as Fig.9 (Measured CE (y) versus Modelled CE (x)).

**As for Fig 10 we prefer this representation because it stretches more the experimental uncertainties.**
**Since the scaling is smaller on the Y-axis than the X-axis, it is better to see the measurement and the corresponding errors on the X-axis. If we switch on the Y-axis the error bar will be smaller.**

Fig.10: Add a second curve to show the modelled collection efficiency without any charge effects so we can see the relative importance for the CE of charge compared with dynamic effects.
**The aim of the Fig 7 and 8 is to observe the influence of the electric charge on the CE. So, you can find the theoretical curves without the electrostatic effects (and the modelled CE with electrostatic effects following your comment).**

**Fig 9 and 10 appears in the section "3.2.3 Comparison with existing models" where we only focus on the comparison of our measurements with the Kraemer and Johnstone prediction (1955) and our model. Thus, we did not add a curve with the modelled CE without electrostatic effects. Moreover, it is the 4 curves for the 4 AP radii (like in Figure 8) which have to be added vertically and horizontally and the figure would have been too crowded.**

Fig.10: Please indicate in the caption what the error bars indicate. They are clearly underestimated and do not represent the full errors for each point. Please indicate the magnitude of the systematic errors either on a few representative points or else quoted in the caption.

**You are right, we underestimated some point at very low CE ($\leq 10^{-4}$). As stated in 3.2.3.2, it is explained by the detected AP mass which is close to the detection limit of the fluorimeter. So, a post experiment pollution was probably involved. The problem is that this pollution is difficult to evaluate even if It can change a lot our measurements. Before your comment, we just omitted this uncertainty but now we did the choice to increase the low uncertainty of the CE measurements smaller than $10^{-4}$.**

**This choice is explained at the end of the new section "2.3 Uncertainties":**

**"As mentioned in 3.2.3.2, an AP pollution independent from the experiment (pollution during the spectrometry analysis, when the droplet impaction cup is extracted at the end of experiments, etc.) remains and should be considered in equation (5). Indeed, it can significantly increase the CE measurement, especially when the measured AP mass is close to the detection limit. It means, when the CE is below $10^{-4}$ for the experiment durations considered in the study. Instead of omitting this uncertainty which is difficult to evaluate, the low uncertainty for the CE measurements below $10^{-4}$ were increased until the end of the axis in Figure 7 and 10. "**

**Note that this choice is reminded in section 3.2.3.2 in the 2$^{nd}$ paragraph which states about a possible AP possible for the CE measurement below $10^{-4}$ :**

**"Note that the detection limit of the spectrometer is $10^{-15}$ kg (for the nominal analysis volume considered), which only represents ten APs with a dry radius of 250 nm deposited on the droplet impaction cup. Thus, it exists an important uncertainty in these CE measurements related to a possible contamination. This is difficult to quantify but the low uncertainties of the CE measurements below $10^{-4}$ were increased in Figure 10. To reduce this potential pollution, it would be necessary to work in a cleanroom or increase the experiment duration to avoid detection problem. However, for these data points the experiment duration was almost 6 hours (without mentioning the preparation, the purging and the cleaning durations) and, beyond this duration, stability problems of the piezoelectric droplet generator were frequent."**

**For the vertical error bar for the modelled CE, it is explained in 3.2.3.1 and 3.2.3.2 :**

**"The horizontal error bars are the measurement uncertainties while the vertical ones are the extreme theoretical CE values considering the extreme droplet and AP charges (by adding or subtracting the charge uncertainties).".**

Fig.10: State the experimental conditions – or their range – in the caption.

**As Fig 8 and 9 we added the droplet radius following your previous comment. In the three captions we added "The experimental conditions are summarised in Table 5." Where the Table 5 give the key features of the experiments. We added this table according to your comment for the part I paper.**

l.525: Replace "got" with "have". **We updated following your comment.**

l.560: Replace "considering" with "that include". **We updated following your comment.**

Fig.12: This is another figure that uses 3D images but would be far clearer and more useful if it were replaced by a simple line schematic. Please indicate precisely where on the new figure the image shown in Fig.13 is obtained.

**We didn't change this figure since we feel the left and mid panels offer different points of view and can be useful for people more comfortable with 3D view. The right panel of this figure looks like a cross section. So, we feel, everyone can understand this *ex situ* experiment by examining the panel of their choice.**